# Oxidation of SQSTM1/p62 mediates the link between redox state and protein homeostasis

Bernadette Carroll[1], Elsje G. Otten[1], Diego Manni[1], Rhoda Stefanatos[1], Fiona M. Menzies[2], Graham R. Smith [1,3], Diana Jurk [1], Niall Kenneth[1], Simon Wilkinson [4], Joao F. Passos[1], Johannes Attems[5], Elizabeth A. Veal[1], Elisa Teyssou[6], Danielle Seilhean[6,7], Stéphanie Millecamps[6], Eeva-Liisa Eskelinen[8], Agnieszka K. Bronowska[9], David C. Rubinsztein[2,10], Alberto Sanz [1] & Viktor I. Korolchuk[1]

Cellular homoeostatic pathways such as macroautophagy (hereinafter autophagy) are regulated by basic mechanisms that are conserved throughout the eukaryotic kingdom. However, it remains poorly understood how these mechanisms further evolved in higher organisms. Here we describe a modification in the autophagy pathway in vertebrates, which promotes its activity in response to oxidative stress. We have identified two oxidation-sensitive cysteine residues in a prototypic autophagy receptor SQSTM1/p62, which allow activation of pro-survival autophagy in stress conditions. The *Drosophila* p62 homologue, Ref(2)P, lacks these oxidation-sensitive cysteine residues and their introduction into the protein increases protein turnover and stress resistance of flies, whereas perturbation of p62 oxidation in humans may result in age-related pathology. We propose that the redox-sensitivity of p62 may have evolved in vertebrates as a mechanism that allows activation of autophagy in response to oxidative stress to maintain cellular homoeostasis and increase cell survival.

[1] Institute for Cell and Molecular Biosciences (ICaMB), Newcastle University Institute for Ageing (NUIA), Newcastle University, Campus for Ageing and Vitality, Newcastle upon Tyne, NE4 5PL, UK. [2] Cambridge Institute for Medical Research, Wellcome Trust/MRC Building, Hills Road, Cambridge, CB2 0XY, UK. [3] Bioinformatics Support Unit (BSU); Faculty of Medical Sciences, Newcastle University, Newcastle Upon Tyne, NE2 4HH, UK. [4] Edinburgh Cancer Research UK Centre, Institute of Genetics and Molecular Medicine, Western General Hospital, University of Edinburgh, Edinburgh, EH4 2XR, UK. [5] Institute of Neuroscience (IoN); Newcastle University Institute for Ageing (NUIA), Newcastle University, Campus for Ageing and Vitality, Newcastle upon Tyne, NE4 5PL, UK. [6] Institut du Cerveau et de la Moelle épinière (ICM), INSERM U1127, CNRS UMR7225, Sorbonne Universités, Université Pierre et Marie Curie, University of Paris 06, UPMC-P6 UMRS1127, Hôpital Pitié-Salpêtrière, Paris, France. [7] Département de Neuropathologie, AP-HP, Hôpital de la Pitié-Salpêtrière, Paris, France. [8] Department of Biosciences, University of Helsinki, Helsinki, 00790, Finland. [9] School of Chemistry, Newcastle University, Newcastle upon Tyne, NE1 7RU, UK. [10] UK Dementia Research Institute, University of Cambridge, Hills Road, Cambridge, CB2 0XY, UK. Bernadette Carroll, Elsje G. Otten, Diego Manni and Rhoda Stefanatos contributed equally to this work. Correspondence and requests for materials should be addressed to V.I.K. (email: viktor.korolchuk@ncl.ac.uk)

Pathological processes such as ageing and age-related diseases are commonly associated with oxidative DNA and protein damage, with cells employing multiple homoeostatic mechanisms to promote survival in these conditions[1,2]. Autophagy has an important role in maintaining a functional proteome by degrading damaged proteins and organelles that are potentially toxic[3]. Autophagy is also important for cell survival in stress conditions[4] and promotes longevity in a diverse range of species from yeast to mammals[5], however it remains unknown whether the longer lifespan characteristic of higher organisms requires evolutionary adaptations to the autophagy machinery[4,6–10].

Recruitment of substrates for degradation through the autophagy pathway is mediated by a family of receptor proteins[11,12]. SQSTM1/p62 is a prototypic autophagic receptor molecule that links ubiquitylated substrates to the nascent autophagic vesicles[11,13]. p62 contains extended disordered regions and several structural domains including an N-terminal PB1 (Phox and Bem1p) domain, which is involved in non-covalent oligomerisation of the molecule[14–19]. The PB1 domain is essential for the formation of intracellular aggregates of p62 and associated ubiquitylated proteins (e.g. in conditions of autophagy deficiency). Oligomerisation and aggregation of p62 is also important for autophagic degradation of the protein together with associated substrates by mediating high-avidity binding to LC3-II on nascent autophagic membranes and promoting the formation of the autophagosome around the cargo[16,17,20].

Age-related neurodegeneration in p62 knockout mice and the identification of sporadic mutations in p62 in patients suffering from late-onset amyotrophic lateral sclerosis (ALS) suggest that p62 is important for neuronal survival[21,22]. Several of the p62 mutations identified in ALS are located within disordered regions of the molecule, and the mechanisms by which they may lead to the disease remain unknown[22–24].

Here we demonstrate that in vertebrates, p62 senses the saturation of reactive oxygen species (ROS)-buffering systems, and that this redox-sensitivity is important to increase autophagy, and thus ensure the survival of cells under oxidative stress conditions. Furthermore, our data suggest one of the mutations causing ALS affects the redox-sensitivity of p62.

## Results

**Oxidation of p62 promotes its oligomerisation.** Our comparative analysis of mouse brain tissue detected an accumulation of high molecular weight species of p62 in old animals (Fig. 1a). These were sensitive to reducing agents suggesting that they are mediated by oxidation and disulphide bond formation involving p62 molecules and therefore designated disulphide-linked conjugates (DLC). Oxidation of p62 was consistent with increased hyperoxidised peroxiredoxins (PRDX-SO$_3$) in old tissues, signifying redox imbalance[25] (Fig. 1a). Despite no observable differences in the levels of p62 mRNA or protein in brain tissue (Fig. 1a; Supplementary Fig. 1a), intracellular aggregates of p62 were significantly increased in neurons of old animals, suggesting that DLC of p62 may contribute to its aggregation (Fig. 1b).

To investigate the mechanism of p62 DLC formation, we exposed cultured cells to a range of treatments and analysed by western blotting in non-reducing and reducing conditions (Fig. 1c; Supplementary Fig. 1b). Treatments with autophagy inhibitors bafilomycin A1 and chloroquine[14,26,27] promoted the accumulation of p62 without significant formation of DLC (Fig. 1c). Likewise, the ribosomal poison puromycin and apoptosis inducing agent staurosporine, which are both known to cause oligomerisation and aggregation of p62[13,28] did not

induce DLC formation (Supplementary Fig. 1b). In contrast, oxidative stress induced by H$_2$O$_2$ or PR-619 (a redox cycler known to produce hydrogen peroxide in aqueous solutions, cause oxidation of cysteine residues and non-specifically inhibit activity of deubiquitylating enzymes (DUBs)[29,30]) resulted in the formation of p62 conjugates similar to those observed in mouse tissue (cf. Fig. 1a, c). Treatment with PR-619 produced more p62 DLC compared to H$_2$O$_2$, with the majority of the protein migrating as high molecular weight complexes including a fraction retained in the stacking gel (Fig. 1c). This correlated with significantly higher levels of intracellular ROS compared to H$_2$O$_2$ treatment indicating that PR-619 is a stronger pro-oxidant (Supplementary Fig. 1c). The ability of PR-619 to induce both increases in intracellular ROS and p62 DLC was suppressed by antioxidants such as N-acetylcysteine (NAC, Supplementary Fig. 1c, d). In contrast, another DUB inhibitor N-ethylmaleimide did not induce the formation of p62 DLC (Supplementary Fig. 1e). Treatment of purified recombinant human p62 expressed in E. coli with H$_2$O$_2$ or PR-619 in vitro induced the formation of DLC confirming the direct pro-oxidant activity of PR-619 (Supplementary Fig. 1f), and importantly, demonstrating that p62 alone is sufficient for the formation of DLC and does not require inhibition of DUBs.

In conditions of p62 DLC formation, we also observed an increase in intracellular p62 aggregates compared with untreated controls (Fig. 1d). Further support for the aggregation-promoting role of DLC came from ultracentrifugation analysis, which showed that induction of DLC shifts p62 into an insoluble fraction (Supplementary Fig. 2a). Therefore, in addition to non-covalent interactions such as PB1 domain-dependent oligomerisation, aggregation of p62 can also be facilitated by DLC formation.

Analysis of the kinetics of p62 oxidation in tissue culture with H$_2$O$_2$ demonstrated a rapid concentration-dependent response (Supplementary Fig. 3a). Formation of p62 DLC appeared to be reversible at lower concentrations of H$_2$O$_2$ with a more sustained response at higher doses. To test whether DLC can be reduced by thioredoxin family proteins we examined the effect of thioredoxin reductase inhibitors curcumin[31] or auranofin[32] on this process. Both treatments induced sustained DLC formation consistent with thioredoxin activity being required to reduce DLC (Supplementary Fig. 3b). These observations suggest that p62 DLC are stabilised when protein oxidation is increased beyond the capacity of thioredoxin to reduce oxidised oligomers. Treatment with PR-619 resulted in a gradual increase in p62 DLC formation with increasing length of cell exposure, suggesting that it resulted in sustained oxidative stress (Supplementary Fig. 3c). Therefore, H$_2$O$_2$ and PR-619 were subsequently used in parallel as tools to stimulate DLC formation.

**Cysteine residues 105 and 113 mediate p62 DLC formation.** To identify which cysteine (Cys) residues were involved in the formation of p62 DLC, four fragments of p62 were analysed (Supplementary Fig. 4a). The N-terminal region (amino acids 1–122), containing the PB1 domain and the flanking unstructured regions, was found to be sufficient to form DLC in response to oxidative stress, whereas the rest of the molecule did not appear to participate in this process (Supplementary Fig. 4b). The 1–122 fragment contains five Cys residues with C105 and C113 being the most conserved among the 14 analysed vertebrate species (Fig. 2a; Supplementary Fig. 5a). The two Cys residues with the highest degree of conservation lie in a disordered region encompassing amino acids 102–121 (Supplementary Fig. 5a). Interestingly, while no single cysteine (C) to alanine (A) substitution completely prevented DLC formation, single

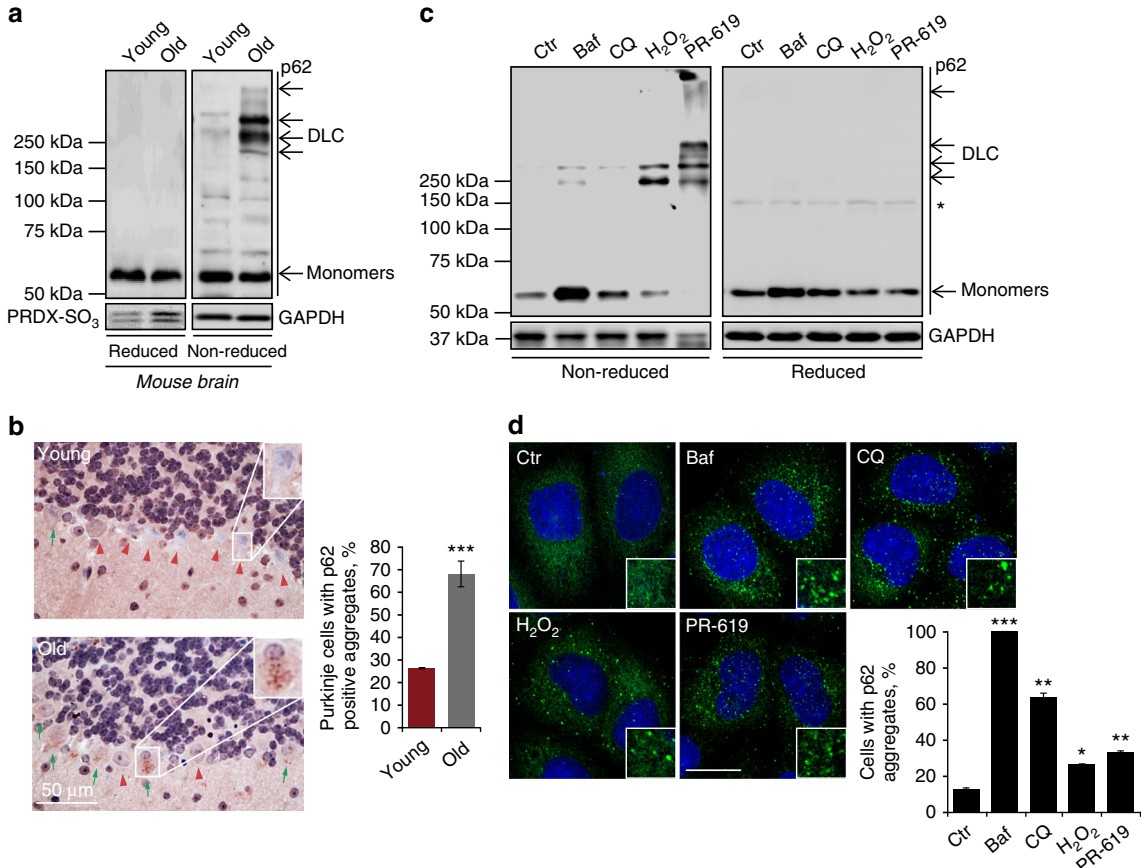

**Fig. 1** p62 forms oligomers and aggregates in response to oxidation. **a** Mouse brain tissue, young (3 months) and old (24 months) analysed by immunoblotting for p62, PRDX-SO3 and actin as a loading control in reducing (2.5% β-ME) and non-reducing conditions. DLC disulphide-linked conjugates. Arrows indicate the positions of monomeric and oligomeric p62. **b** Representative images and quantification of p62 aggregates in old mouse Purkinje cells in the cerebellum. Green arrows and red arrowheads indicate Purkinje cells positive and negative for p62 aggregates, respectively. **c** Effect of autophagy inhibition (bafilomycin A1 (Baf, 400 nM, 4 h) and chloroquine (CQ, 50 μM, 4 h)) and oxidative stress ($H_2O_2$ (3 mM, 10 min) and PR-619 (5 μM, 30 min)) on p62 DLC (**c**) and p62 aggregation (**d**) in HeLa cells. **d** Anti-p62 staining analysed by confocal microscopy. Error bars represent s.e.m., $n = 3$, *$P < 0.05$, **$P < 0.01$, ***$P < 0.005$ (unpaired $t$-tests). Asterisk indicates a non-specific band. Scale bar: 20 μm

mutagenesis of either C105 or C113 resulted in a partial impairment of this process (Supplementary Fig. 5b). Moreover, double substitution of C105 and C113 significantly reduced the levels of p62 oxidation detected after $H_2O_2$ or PR-619 treatments in cell culture and in vitro (Fig. 2b; Supplementary Fig. 5c, d). Substitutions of other Cys residues did not further ablate DLC formation (Supplementary Fig. 6), suggesting that C105 and C113 residues are required for efficient ROS sensing. The K7A,D69A double-mutation, which disrupts the PB1 domain-dependent oligomerisation of p62[14], did not suppress DLC formation (Supplementary Fig. 5c) suggesting that the formation of disulphide bonds does not require prior non-covalent interactions via the PB1 domain.

Notably, the C105A,C113A mutant formed significantly fewer intracellular aggregates following exposure to $H_2O_2$ or PR-619, consistent with DLC formation promoting the aggregation of p62 (Fig. 2c, d). Furthermore, ultracentrifugation experiments confirmed that wild type, but to a much lesser extent C105A, C113A-mutant p62, was redistributed to the insoluble fraction upon exposure to oxidants (Supplementary Fig. 7a, b). In agreement with the requirement of p62 for the formation of ubiquitylated protein aggregates[33,34], p62 aggregates were ubiquitylated and as such cells expressing C105A,C113A p62 also displayed significantly fewer intracellular ubiquitin-positive aggregates (Fig. 2c, d).

As both PB1 domain-dependent and disulphide bond-mediated mechanisms appear to contribute to p62 aggregation, we investigated the relationship between the two processes. Although disruption of PB1 domain-mediated interactions (K7A, D69A mutation) did not preclude the formation of DLC in response to oxidation (Supplementary Fig. 5c), it did suppress the formation of intracellular p62 aggregates in the same conditions (Supplementary Fig. 8a). Thus, although oxidation of p62 can promote p62 oligomerisation, the PB1 domain appears to be required for sequestration of these oligomers into microscopically observable aggregates. Although the PB1 domain is also essential for the formation of p62 aggregates in conditions of autophagy inhibition (Supplementary Fig. 8a), C105/C113 residues were dispensable for this process (Supplementary Fig. 8b, c) suggesting that redox sensing by p62 is important for its aggregation specifically in response to oxidative stress.

**Oxidation of p62 activates pro-survival autophagy.** Cycloheximide chase experiments demonstrated slower degradation of the C105A,C113A mutant compared to wild-type p62 in oxidative stress conditions, which was further confirmed using the Click-iT pulse-chase technique (Fig. 3a; Supplementary Fig. 9a, b). The difference between the C105A,C113A-mutant and wild-type p62 was cancelled out by bafilomycin A1 or

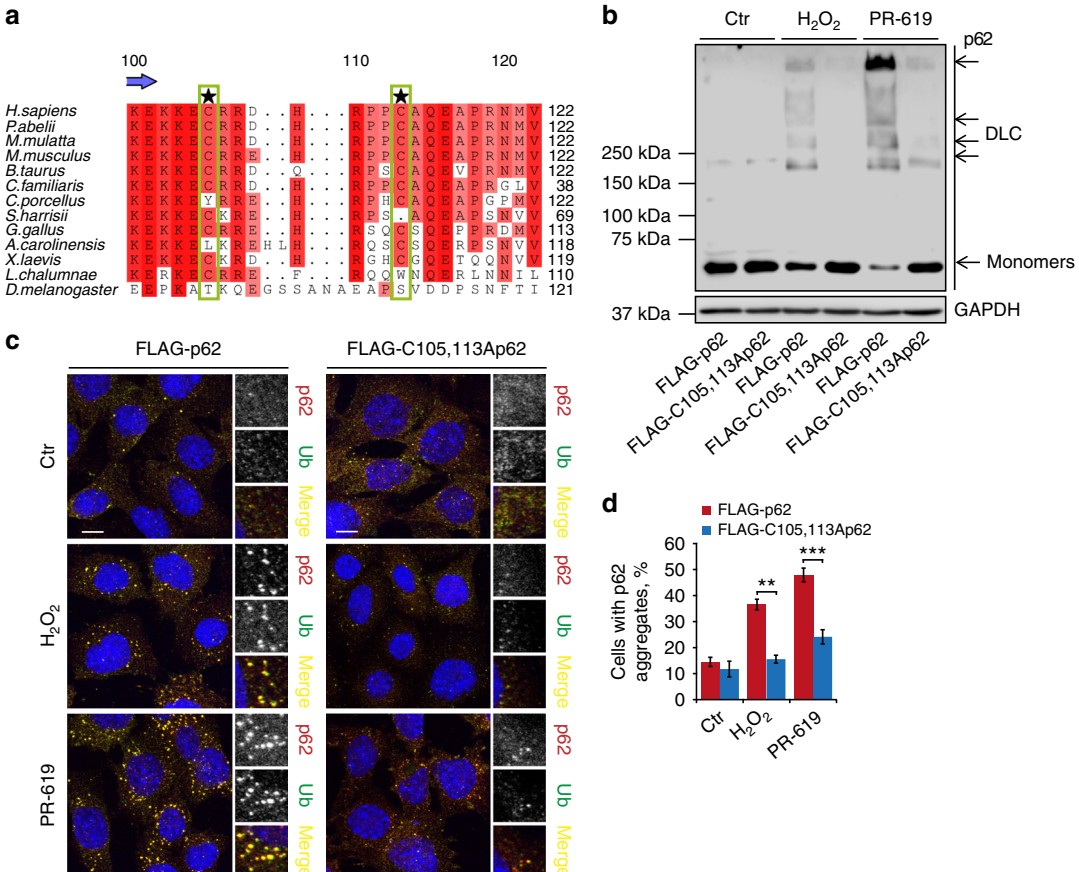

**Fig. 2** Two conserved cysteine residues located in a disordered region of p62 are required for formation of DLC and ubiquitylated aggregates. **a** Alignment showing cysteines 105 and 113 (highlighted) are conserved in vertebrates. Increasing conservation across species is shown by light-to-dark red. **b** $p62^{-/-}$ MEFs stably expressing either wild type or C105A,C113A FLAG-p62 were treated with $H_2O_2$ (3 mM, 1 min) or PR-619 (5 μM, 10 min) and immunoblotted in non-reducing conditions. **c** Formation of ubiquitylated, p62-positive aggregates in stable cells described in **b** following oxidative stress (1 mM $H_2O_2$ or 5 μM PR-619 for 30 min). Cells were immunostained for ubiquitin and p62, analysed by confocal microscopy (**c**) and quantified (**d**). Error bars represent s.e.m., $n = 3$, **$P < 0.01$, ***$P < 0.005$ (unpaired $t$-tests). Arrows indicate the positions of monomeric and oligomeric p62. Scale bar: 10 μm

chloroquine, suggesting that oxidation of p62 promotes its degradation by autophagy (Fig. 3b; Supplementary Fig. 9c). Indeed, wild type but not the C105A,C113A mutant of p62 appeared to stimulate autophagy as indicated by increased levels of LC3-II and reduced levels of ubiquitylated proteins and p62 itself (Fig. 3c, d). This effect on autophagy markers was particularly pronounced upon exposure to $H_2O_2$ or PR-619, suggesting the importance of p62 for oxidative stress-induced autophagy (Fig. 3c, d; Supplementary Fig. 9d). The failure of p62-null cells or cells expressing oxidation-insensitive p62 mutant to induce autophagy was also evident from the significantly reduced numbers of autophagosomes as assessed by immuno-fluorescence (Fig. 3e, f) and electron microscopy (Fig. 3g, h). Oligomerisation of p62 via non-covalent PB1 domain-dependent interactions has previously been suggested to promote the formation of the autophagosome around cargo[16,17]. Cells expressing the PB1 domain mutant (K7A,D69A) of p62 displayed a similar deficiency in autophagy to that observed in C105A,C113A-mutant cells (Fig. 3c–h; Supplementary Fig. 9d), suggesting that common mechanisms underlie the role of non-covalent and disulphide bond-mediated p62 oligomerisation in promoting autophagosome biogenesis.

As degradation of misfolded proteins by autophagy has been shown to be important for cell survival[35,36], we investigated whether oxidation of p62 promotes cell viability in oxidative stress conditions. Strikingly, compared to cells expressing wild-type FLAG-p62, cells lacking p62 or expressing the C105A,C113A mutant were more susceptible to cell death following oxidative stress (Fig. 3i, j; Supplementary Fig. 9e)[37,38]. Likewise, the PB1 domain mutant failed to maintain cell viability, in agreement with a common underlying mechanism of p62 oligomerisation as a pre-requisite for cell survival (Fig. 3i, j). The differences in cell survival cannot be explained by the effect of p62 on oxidative stress response via Nfr2 (Supplementary Fig. 10a–d)[39]. Instead, bafilomycin A1 or chloroquine treatment cancelled out the difference in cell survival between the cell lines, confirming autophagy as a mechanism of resistance (Fig. 3j). On the basis of this data, we propose that p62 oxidation and oligomerisation promotes autophagosome biogenesis and degradation of ubiquitylated autophagy cargo and is therefore important for maintaining cell viability in stress conditions.

**Oxidation of p62 confers stress resistance in vivo.** Interestingly, although mouse and human p62 is oxidised (Fig. 1), the oxidation-sensitive Cys residues are not conserved in p62 homologues in invertebrates including flies and worms (Supplementary Fig. 5a). Indeed, the *Drosophila* homologue, Ref(2)P does not form DLC in old animals or in response to $H_2O_2$ (Fig. 4a; Supplementary Fig. 11a). This raises the possibility that this lack of redox-sensitivity is responsible for the observed accumulation of Ref(2)P in aged flies (cf. Figs. 1a and 4a). To test

whether acquisition of ROS sensing by p62 might confer an advantage by promoting autophagy and stress resistance, an eighteen amino acid fragment of human p62 containing C105 and C113 was used to replace the corresponding region in the fly Ref(2)P gene using CRISPR/Cas9 technology (Fig. 4b). The

resulting protein was shown to be oxidised in response to $H_2O_2$ and therefore designated Ref(2)P$^{OX}$ (Fig. 4b; Supplementary Fig. 11a). Compared to wild type, Ref(2)P$^{OX}$ flies displayed increased autophagy flux, as indicated by the increased levels of LC3-II homologue Atg8-II in the presence of the lysosomal

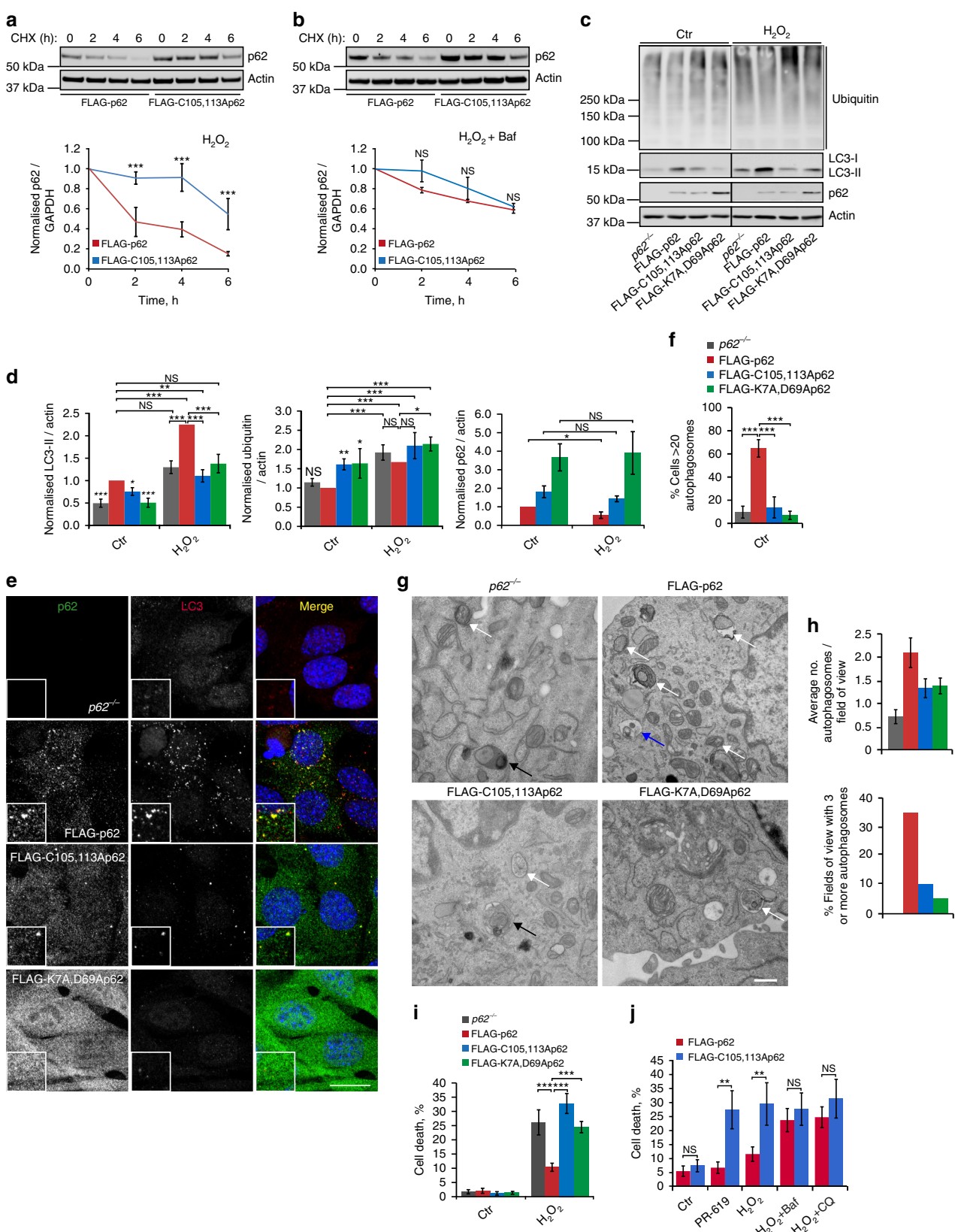

inhibitor chloroquine (Supplementary Fig. 11b). No differences in Atg8-II levels were detected in the absence of lysosomal inhibitor, likely to be due to its increased turnover together with ubiquitylated proteins and Ref(2)P$^{OX}$ in aged Ref(2)P$^{OX}$ animals (Supplementary Fig. 11c). Strikingly, in acute stress conditions, induced by paraquat or thermal stress[40,41], the survival of Ref(2)P$^{OX}$ flies was extended, which also correlated with enhanced turnover of ubiquitylated proteins and Ref(2)P$^{OX}$ itself (Fig. 4c, d; Supplementary Figs. 11d, 12a, b).

To investigate whether perturbation of p62 oxidation could underlie pathology in humans, we searched for known mutations in the highly flexible loop of p62 containing the C105/C113 residues. A K102E mutation (Supplementary Fig. 5a) has previously been identified as causative in sporadic ALS (SALS)[23,24]. On the basis of the above data, we generated a hypothetical in silico model of disulphide bond-mediated p62 oligomerisation, which predicted that K102 may be involved in the salt-bridge formation with E19 within the PB1 domain (Supplementary Fig. 13a). This salt bridge could stabilise the protein, and also affect the conformation and orientation of the near cysteine C105, enabling formation of the disulphide bond between C105 and C113 from another p62 subunit. K102E mutation would disrupt this stabilising interaction and also perturb oxidation of the redox-sensitive Cys105 residue through a large conformational change of the loop, making it inaccessible for the formation of further disulphide bonds (Supplementary Fig. 13a). Indeed, we found that the K102E mutation (but not a number of unrelated mutations within this region) impaired the ability of p62 to form DLC in response to oxidative stress (Fig. 4e; Supplementary Fig. 13b), as well as its ability to induce autophagy, thus suggesting a pathological mechanism for this mutation in SALS (Supplementary Fig. 13c, d). Determining the nature of p62 and DLC in ALS patients harbouring the K102E mutation will be an important future step to validate this model.

## Discussion

The formation of p62 oligomers in response to oxidative stressors, such as $H_2O_2$[42], cigarette smoke[43] and UV-irradiation[44] has been previously reported. However, the high molecular weight species of p62 observed were resistant to reducing agents suggesting a separate formation mechanism to that described here. The Cys105/113 residues required for DLC formation, together with Lys102 (which is mutated in SALS) locate to a previously described charged bridge, which was suggested to align in the antiparallel orientation and promote oligomerisation of p62 via non-covalent PB1 domain-mediated interactions[45]. Interestingly, although the formation of intermolecular disulphide bonds between oxidation-sensitive Cys residues in this model is possible, the resulting DLC will be limited to p62 dimers that are not observed in our experiments. Instead, we propose a putative model of DLC formation involving a parallel orientation of p62 molecules, which could also explain the pathological mechanism of the K102E mutation (Supplementary Fig. 13a). Although this

model requires testing in future structural studies our mutational analyses suggest that the two processes of PB1 domain- and DLC-mediated oligomerisation can act relatively independently, however both can contribute to the formation of p62 aggregates. Importantly, DLC formation appears to initiate or facilitate p62 oligomerisation in conditions of oxidative stress and promote the formation of microscopically observable aggregates, the latter process also requiring the function of its PB1 domain. Importantly, cells carrying oxidation-insensitive or PB1 domain defective mutants of p62 have similar functional phenotypes (e.g. autophagy and cell survival defects) suggesting that, regardless of the underlying molecular mechanism, oligomerisation of p62 is required for its cellular function.

Overall, our data implies that oxidation-dependent oligomerisation of p62 promotes autophagy and is a pro-survival mechanism. In line with previous observations[16,17,20,46], our data show that autophagy receptors such as p62 act not only as cargo carriers but also stimulate the process of autophagosome formation. Furthermore, we propose that the sensitivity of p62 to ROS-induced oxidation may have evolved in vertebrates as a mechanism that allows more efficient autophagy during oxidative stress conditions, thereby maintaining cellular homoeostasis and increasing cell survival (Fig. 4f). This mechanism may be particularly relevant to longer-lived species with a high metabolic rate such as humans where it counteracts the effect of age-associated oxidative stress and promotes survival of differentiated cells, in particular neurons, for years and even decades of human lifespan. Consistent with this view, our data suggest that perturbation (e.g. by K102E mutation) of this redox sensitivity may occur in age-related pathology in humans, although this remains to be formally tested in vivo (Fig. 4f). In recent years, a growing number of signalling proteins have been shown to be regulated by ROS-induced changes in the redox state. Our data suggest that such changes may also have an unanticipated role as a mechanism of aggregation and degradation of proteins via the autophagy pathway thus promoting cellular protein homoeostasis in oxidative stress conditions. This has wider implications for the myriad of important processes in which autophagy is implicated including ageing, cancer cell survival and ischaemia/reperfusion injury.

## Methods

**Cell lines**. HeLa and HEK293E cells were obtained from ECACC (European Collection of Cell Cultures). p62 knockout ($p62^{-/-}$) and wild-type ($p62^{+/+}$) mouse embryonic fibroblasts (MEFs) were kindly provided by Eiji Warabi of the University of Tsukuba. HEK293FT lentivirus packaging cells were from Invitrogen (Paisley, UK). Cells were grown in DMEM (Dulbecco's Modified Eagle's Medium, Sigma #D6546) supplemented with 10% heat-inactivated FCS (Foetal Calf Serum, Biosera), 5% penicillin/streptomycin (Invitrogen) and 2mM L-glutamine (Sigma) in a humidified atmosphere containing 5% $CO_2$ at 37 °C. Cells were treated with puromycin (Santa Cruz Biotechnology), staurosporine (Sigma), z-VAD (Sigma), $H_2O_2$ (Sigma; the $H_2O_2$ stock was replaced every 2 weeks for the duration of the project), PR-619 (LifeSensors), curcumin (Sigma), auranofin (Sigma), N-acetylcysteine (Sigma), bafilomycin A1 (Enzo Life Sciences), chloroquine (Sigma), cycloheximide (Sigma) and retinoic acid (as an inhibitor of Nrf2[47]) at different

**Fig. 3** Oxidation-sensitive p62 is required for pro-survival autophagy. **a** $p62^{-/-}$ MEFs stably expressing FLAG-tagged wild type or C105A,C113A p62 were treated with cycloheximide (CHX, 50 µg/ml) and $H_2O_2$ (1 mM), either in the absence (**a**) or presence (**b**) of bafilomycin A1 (Baf, 50 nM), lysed at the indicated time post treatment, immunoblotted for p62 and quantified. **c** Cells described in **a** plus one stably expressing K7A,D69A PB1-domain mutant of p62 were treated with $H_2O_2$ (1 mM, 5 h), lysed and immunoblotted for ubiquitin, LC3, p62 and actin (**c**) and quantified (**d**). **e, f** Stable p62 cell lines in control conditions were fixed and stained for p62 and LC3 (**e**) and the % cells with >20 autophagosomes was quantified (**f**). **g, h** Electron microscopy of stable p62 cell lines. White arrows: autophagosomes; black arrows: autolysosomes; blue arrow: endosome. Scale bar: 500 nm (**g**). Autophagosomes were quantified and graphs represent average number and % cells with three or more autophagosomes per field of view (**h**). **i, j** Stable p62 cell lines were treated as in **c** and % cell death was analysed by Ready Probes fluorescent dyes (**i**). **j** Stable p62 cell lines were treated as indicated and % cell death was analysed as in **i**. Error bars represent s.e.m., n = 3, *P < 0.05, **P < 0.01, ***P < 0.005 (unpaired t-tests). NS not significant. Scale bar: 20 µm (IF) and 500 nm (TEM)

concentration and time-points as indicated. Medium was switched to serum-free DMEM for the duration of the treatments.

**Transfection.** HeLa cells and MEFs were seeded in either 6- or 12-well plates, cultured for 24 (HeLa) or 48 (MEFs) hours and transfected with Lipofectamine 2000 (Invitrogen) according to the manufacturer's instructions for 24 or 48 h prior to lysis. Plasmids used in this study: pEGFP-p62, pEGFP-ΔUBAp62 and pDEST-GST-p62 were kindly provided by Terje Johansen (University of Tromsø, Tromsø,

Norway)[14], FLAG-p62 was kindly provided by Robert Layfield (University of Nottingham, Nottingham, UK)[48], pEGFP-C2 (Clontech, #632481) and pLENTI6/V5-DEST (Invitrogen, #V496-10).

**Mutagenesis.** Point mutagenesis of p62 was carried out using the QuikChange II XL Site-Directed Mutagenesis Kit (Agilent Technologies) following the provided protocol. Mutagenesis primers were designed using the QuikChange Primer Design Program available online on the Agilent website (Supplementary Table 1). The

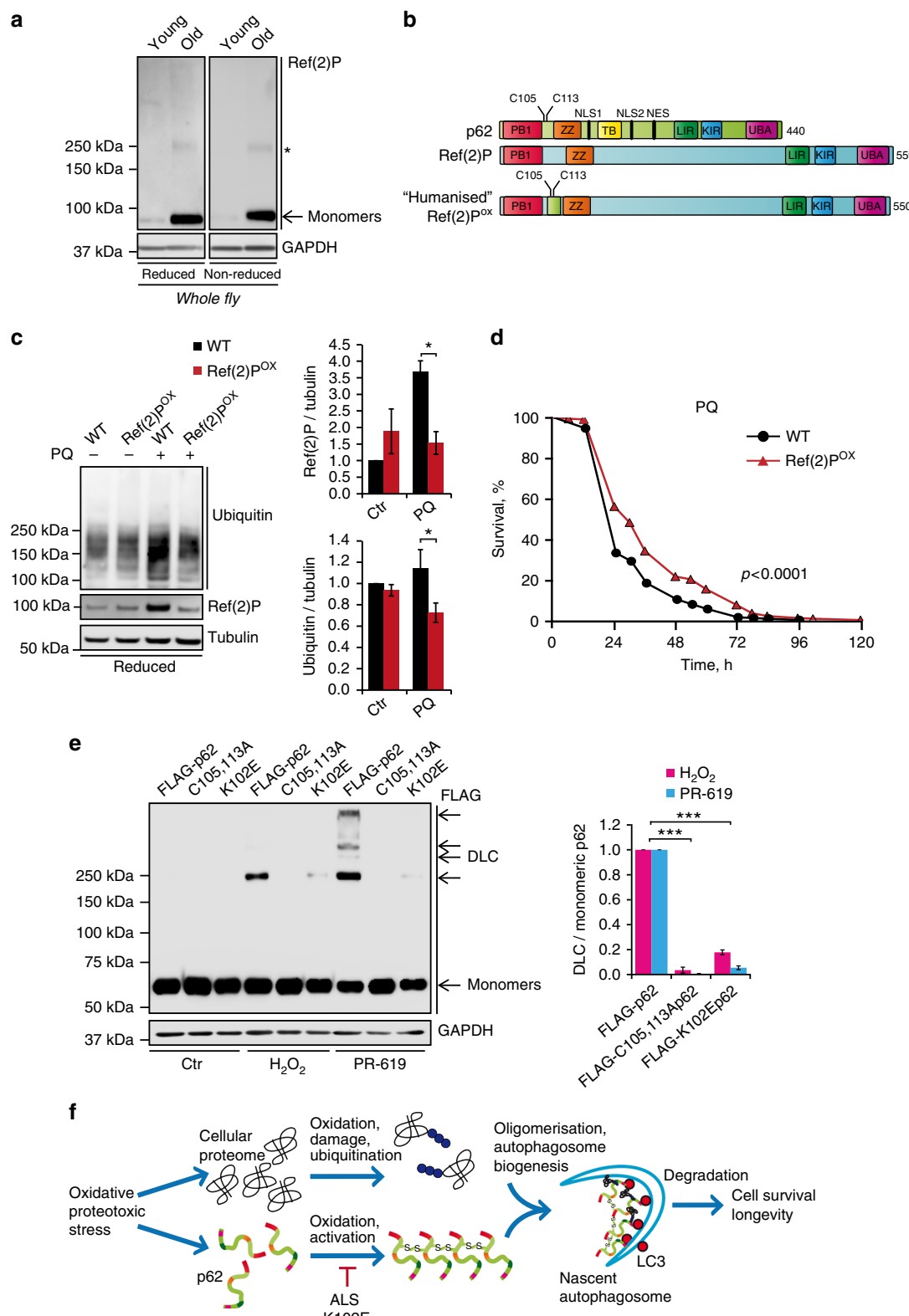

pEGFP-p62, Flag-p62 and GST-p62 wild-type plasmids were used as a template. PCR reactions were run on a Veriti 96-Well Thermal Cycler (Applied Biosystems). Transformation of bacteria was then carried out using XL10-Gold ultracompetent cells provided with the kit. Bacterial DNA was extracted using the QIAprep Spin Miniprep Kit (Qiagen) and sent for sequencing (Genevision, Newcastle Upon Tyne, UK).

**Cloning**. Deletions of p62 were produced through PCR, using the pEGFP-p62 wild type as a template and specific primers generated by SnapGene software (GSL Biotech) (Supplementary Table 1). The generated fragments were then ligated into the pEGFP-C2 vector. Forward primers included a *Bgl*II restriction site. PCRs were performed using Phusion High-Fidelity DNA polymerase (Thermo Scientific) following the PCR standard protocol. PCR reactions were run on a Veriti 96-Well Thermal Cycler (Applied Biosystems). p62 fragments were then digested with *Bgl*II (sticky end, NEB) and run on agarose gel which was then purified. The pEGFP-C2 vector was digested with *Sma*I (blunt end, NEB) and run on agarose gel followed by purification with QIAquik GEL Extraction Kit (Qiagen). The pEGFP-C2 vector was then digested with *Bgl*II, followed by gel purification. After double digestion, the vector was dephosphorylated by calf intestinal alkaline phosphatase (Fermentas). The digested p62 fragments were ligated into the double-digested pEGFP-C2 vector (sticky end/blunt-end ligation) using the T4 DNA Ligase (New England BioLabs). Bacterial transformation was performed as described above using α-select GOLD Efficiency chemically competent cells (Bioline). FLAG-p62 and FLAG-C105AC113Ap62 for lentiviral expression were subcloned into the pLENTI6/V5-DEST vector using *Eco*RI and *Xho*I (NEB) as described above.

**qPCR of p62 and Nrf2 targets in MEFs and brain tissue**. Total RNA was extracted in triplicate from MEFs using Trizol (Thermo Scientific) and DNAase treated. cDNA was synthesised using the High-Capacity cDNA reverse transcription kit (Applied Biosystems) according to manufacturers instructions and analysed using an Applied Biosystems Step One qPCR machine. Relative expression of the target genes was normalised to actin levels. Data were extracted and analysed using Applied Biosystems 7500 software version 2.0 and Prism software. For each target mean fold change with standard error is presented.

RNA from mouse tissue was isolated from snap frozen tissues ($n = 5$ per group) using RNeasy Mini Kit and QIAshredder (Qiagen) according to manufacturers instructions. Total RNA was reverse-transcribed using Omniscript Reverse Transcription Kit (Qiagen) according to manufacturers instructions. Samples were analysed in triplicates using Power SyberR Green PCR Master Mix (Invitrogen) in a C1000TM Thermal Cycler, CFX96TM Real-Time System (Bio-Rad) using Bio-Rad CXF Manager software. Expression was normalised to Pkg1. The sequences of all the primers used are included in Supplementary Table 2.

**GST-p62 fusion protein purification**. GST-p62 fusion protein was produced in *E. coli* EXPRESS BL21 (DE3) Chemically Competent Cells (Lucigen) using the pDEST-GST-p62 plasmid[14]. In brief, fusion protein expression was induced by adding 0.5 mM ZnSO4 and 0.05 mM IPTG to induce expression for 3.5 h at 20 °C, Bacterial pellets were lysed by sonication in 50 mM Tris, 150 mM NaCl buffer, pH 7.5 and GST-fusion protein purified using Glutathione-Sepharose 4B beads (GE Healthcare). Purified GST-p62 fusion protein was then treated with $H_2O_2$ and PR-619 as indicated and subjected to immunoblot analysis. Samples were sonicated for 10 s prior to running on an 8% Tris-Glycine SDS-PAGE gel.

**Lentiviral transduction**. Stable expression of FLAG-p62, FLAG-C105A-C113A p62, FLAG-K7A,D69Ap62 and FLAG-K102E p62 in $p62^{-/-}$ MEFs was achieved through lentiviral transduction. Transgenes were cloned into the pLenti6/V5-DEST expression vector containing the blasticidin resistance gene. HEK293FT cells were seeded in antibiotic-free medium supplemented with 0.1 mM MEM non-essential amino acids (Invitrogen) and then cotransfected with either empty or p62 lentiviral expression vectors and 3rd generation packaging system plasmids (Invitrogen). After 24 h, media was replaced with fresh media without antibiotics. 48 h after

transfection, viral transduction was performed by transferring media from HEK293FT cells 70% confluent $p62^{-/-}$ MEFs in the presence of 6 μg/ml Polybrene (Sigma). Media containing virus was replaced after 24 h with fresh media containing 8 μg/ml of blasticidin (Invitrogen) for selection of transduced cells. Media was replaced every 2–3 days for 10–12 days by keeping the antibiotic selection. Transduced MEFs were then maintained in lower levels of blasticidin (4 μg/ml) until seeding for experimental purposes.

**Immunoblot analysis**. HeLa cells, MEFs or HEK293E were seeded in 6- or 12-well plates 24 (HeLa) or 48 (MEFs) hours prior treatments. After treatments, cells were washed in ice-cold 1× PBS then lysed in RIPA buffer (150 mM NaCl, 1% NP40, 0.5% NaDoC, 0.1% SDS, 50 mM Tris pH 7.4, supplemented with 1× Halt Protease & Phosphatase inhibitor cocktail (Thermo Scientific) in ddH2O) plus 50 mM *N*-ethylmaleimide, which interacts with reduced cysteines and prevents new disulphide bond formation during lysis procedures and simultaneously inactivates deubiquitylating enzymes. Cell lysates were then centrifuged at 4 °C at 13,000 rpm for 10 min to remove insoluble cellular components. Protein concentration was measured by Bradford assay using the DC Protein Assay (BioRad) and a FLUOstar Omega plate reader (BMG Labtech). Samples were prepared by boiling in SDS-Loading buffer (BioRad) at 100 °C for 5 min in the presence or absence of 2.5% β-ME (β-mercaptoethanol, Sigma). An amount of 20–40 μg of proteins was run on 10–12% Tris-Glycine SDS-PAGE gels and transferred to Immobilon-P (Millipore) PVDF membranes using a Trans-Blot SD Semi-Dry Transfer Cell (BioRad). Blots were incubated with a blocking solution (PBS containing 5% fat-free dry milk, 0.1% Tween-20) for 1 h. After washing with PBS, blots were incubated with primary antibodies diluted in blocking solution at 4°C overnight. Blots were then washed three times, 5 min each: one time with PBS, once in PBS with 0.1% Tween-20 and one more time in PBS. Then, blots were incubated with secondary HRP-conjugated antibodies α-mouse or α-rabbit (Sigma, #A2554 and #A0545, 1:5000) or α-guinea pig (Dako, #P0141, 1:1000) in blocking solution for 1 h at room temperature. Blots were then washed three times as above. Blots were incubated for 5 min with the Clarity Western ECL Substrate (BioRad) and signals were detected by chemiluminescence using a LAS-4000 CCD camera system (Fujifilm). The following primary antibodies were used: guinea pig α-p62 (Progen Biotechnik #GP62-C, 1:2000), mouse α-GFP (Santa Cruz Biotechnology #sc-9996, 1:1000), mouse α-Ubiquitin (LifeSensors #VU101, 1:1000), rabbit α-PRDX3[49] (1:1000), α-PRDX-SO3 (Abcam #ab16830, 1:2000), mouse α-LC3 (Enzo Life Sciences #ALX-803-081-C100, 1:1000), mouse α-FLAG (Sigma #F1804, 1:2000), rabbit α-FLAG (Sigma #F7425, 1:2000), rabbit α-GAPDH (CST #5174, 1:10,000), rabbit α-actin (CST #4970, 1:2000), rabbit α-*Drosophila* tubulin (Abcam #ab179513, 1:2000), Ref(2)P (Abcam #ab178440, 1:1000) or ([50], a kind gift from Ioannis Nezis, Warwick University, 1:500), α-*Drosophila* Atg8[51] (a kind gift from Ivana Bjedov, UCL, 1:1000), rabbit α-Nrf2 (CST #12721, 1:500) and rabbit α-Histone H3 antibody (CST #9715, 1:2000). Uncropped versions of western blots are presented in Supplementary Fig. 14.

**Ultracentrifugation**. Lysates from HeLa cells and MEFs were first centrifuged at 4 °C at 13,000 rpm for 10 min, and whole-cell samples were prepared by boiling in SDS-Loading buffer (BioRad) at 100 °C for 5 min Supernatants where loaded into 1 ml polycarbonate cuvettes and centrifuged using an Optima TLX Ultracentrifuge (Beckman Coulter, UK) at 100,000 rpm at 4 °C for 1 h. Soluble and insoluble fraction samples were prepared by boiling supernatants and pellets in SDS-Loading buffer (BioRad) at 100 °C for 5 min.

**Cycloheximide degradation assays**. $p62^{-/-}$ MEFs stably expressing FLAG-tagged wild type or C105A,C113A p62 were seeded in 12-well plate until confluent. Cells were treated with cycloheximide (50 μg/ml) and either $H_2O_2$ (1 mM) or PR-619 (5 μM) for 0, 2, 4 and 6 h in serum-free media before being lysed and analysed by immunoblot to monitor p62 degradation. Where co-treatment was carried out with chloroquine (20 μM) or bafilomycin A1 (50 nM), cells were pretreated with these drugs for one hour prior to addition of cycloheximide.

---

**Fig. 4** Oxidation-sensitive p62 is important for the oxidative stress resistance of flies and is perturbed in human age-related disease. **a** Whole-fly lysates were analysed in reducing (2.5% β-ME) and non-reducing conditions for p62 homologue, Ref(2)P. Asterisk indicates a non-specific band. **b** Diagram representing the introduction of an 18 amino acid fragment of human p62 containing C105 and C113 to produce a 'humanised' Ref(2)P (Ref(2)P^ox) in flies using CRISPR/Cas9. **c** Wild-type (WT) and Ref(2)P^ox flies were treated with paraquat (PQ, 20 mM) for 12 h and whole-fly lysates were analysed by immunoblot for ubiquitin, Ref(2)P and tubulin and quantified. Error bars represent s.e.m., $n = 3$ (at least 10 flies per group per replicate); *$P < 0.05$ (unpaired *t*-test). **d** Combined survival data (of three repeats) of wild-type versus Ref(2)P^ox flies in the presence of 20 mM PQ. Survival was assessed every 12 h. At least 60 flies per group per replicate were used and log-rank statistics applied. **e** $p62^{-/-}$ MEFs stably expressing FLAG-tagged wild-type p62, C105A,C113A p62 or ALS-associated mutation K102E p62 were treated with $H_2O_2$ (500 μM, 1 min) and PR-619 (20 μM, 10 min), analysed for p62 DLC formation in non-reducing conditions and quantified. Error bars represent s.e.m., $n = 3$; ***$P < 0.005$ (unpaired *t*-tests). **f** Diagram of the proposed role for p62 oxidation in the aggregation and degradation of autophagy substrates. Formation of p62 DLC is triggered by oxidative stress, which promotes degradation of p62 and bound substrates (e.g. ubiquitylated proteins) through autophagy. Mutations in p62 sequence (e.g. ALS-related K102E) can impair the formation of DLC

**Click-iT AHA assay to measure rates protein degradation**. Cells were seeded into 100 mm dishes and grown until 70–80% confluency. For labelling of newly synthesised proteins, cells were washed once with PBS and incubated overnight with 4 μM Click-iT AHA (L-azidohomoalanine) (ThermoFisher Scientific #C10102) in DMEM without cysteine and methionine (ThermoFisher Scientific #21013024). The following day, AHA was removed and cells were washed with PBS and incubated in serum-free media containing 1mM $H_2O_2$ for the required chase time (0–6 h). Cells were harvested into 400 μl p62 IP buffer (20 mM Tris pH 7.4, 150mM NaCl, 0.5% NP40, 2 mM $MgCl_2$ and protease inhibitor cocktail (Roche)) and incubated on ice for 10 min. Lysates were centrifuged for 10 min at 15,000×$g$, the supernatant was incubated with 20 μl of prewashed Ez-view M2 agarose beads (Sigma) for 2 h at 4 °C with constant rotation. Beads were then washed three times in p62 IP buffer. The AHA containing proteins were then labelled with biotin (biotin alkyne, ThermoFisher Scientific B10185) on the beads via Click-iT Cell Reaction Buffer Kit (ThermoFisher Scientific) and following the manufacturer's protocol. Finally beads were washed three times in p62 IP buffer before protein was eluted by boiling in Laemmli sample buffer (BioRad). Eluted samples were run on 10% SDS-PAGE gels and transferred to low flouresence PVDF membrane (Millipore). Western blots were probed with streptavidin conjugated to IRdye800 (LiCor Biosciences) and visualised using an Odyssey Scanner (LiCor Biosciences). Band intensity was quantified using Image Studio software (LiCor Biosciences).

**Immunofluorescence**. HeLa cells and MEFs were seeded on coverslips in 12-well plates. After treatments, cells were fixed in 4% formaldehyde in PBS for 10 min at room temperature. Cells were permeabilised with 0.5% Triton X-100 for 5 min at room temperature (or in methanol at −20 °C for LC3 staining). Coverslips were blocked for one hour in 5% normal goat or rabbit serum in PBS 0.05% Tween-20 (for LC3 staining, the Tween-20 was omitted from all steps). Cells were incubated with primary antibodies overnight at 4 °C. Primary antibodies used in this study include guinea pig α-p62 (Progen, 1:200), mouse α-p62 (BD Bioscience, 1:200), goat α-Ubiquitin (Santa Cruz Biotechnology #sc-34870, 1:200), rabbit α-LC3 (CST #3868, 1:200). Cells were washed three times and incubated with the appropriate secondary antibodies for 1 h at room temperature (Life Technologies, 1:5000). Cells were washed and nuclear DNA was stained by incubation with TO-PRO-3 iodide (Life Technologies, 1:3000) for 10 min at room temperature. Coverslips were mounted on slides with Prolong Gold Antifade (Life Technologies) and imaged with an LSM 510 META Confocal Microscope (Zeiss) using a 63× Plan-Apo/1.4 NA Oil objective. Images were analysed using ImageJ software (NIH).

**Electron microscopy**. Cells were seeded 48 h prior to treatment and subjected to 2 h amino acid starvation. Cells were collected and fixed overnight in 2% glutaraldehyde in 0.1 M cacodylate buffer. After rinsing in buffer, the cells were post-fixed in 1% osmium tetroxide +1.5% potassium ferricyanide, rinsed in deionised water then dehydrated through a graded series of acetone. Cells were infiltrated with epoxy resin (TAAB medium) and polymerised at 60 °C for 36 h. Ultrathin sections (70 nm) were picked up on copper grids and stained with uranyl acetate and lead citrate before being viewed on a 100kV CM100 TEM (FEI). Images of 20 cells per cell line were collected and quantified.

**Cell death assay**. Cells were seeded 48 h prior to treatments in 12-well plate. Cell viability was assessed using Ready Probes (Life sciences) as per company instructions. Cells were imaged on inverted DM IL LED Leica microscope equipped with an Invenio 3SII digital camera (3.0 Mpix Colour CMOS; Indigo Scientific). Images were analysed using ImageJ and the percentage of cell death was quantified. Please note that to increase the clarity of the final images, cells labelled with NucGreen Live dye have been marked as yellow using ImageJ.

**Nuclear fractionation**. Stably expressing p62 cell lines were seeded in 6-well plates 48 h prior to collection. Cells were washed in ice-cold PBS, scrapped in 1 ml ice-cold PBS and centrifuged for 10 s at 13,000 rpm at 4 °C. The pellet was carefully resuspended in 1 ml ice-cold 0.1% NP40 in PBS. Overall, 50 μl was collected in a fresh tube (whole-cell sample). Samples were centrifuged again for 10 s at 13,000 rpm at 4 °C and supernatant, which represents the cytoplasmic fraction, was discarded. The pellet was resuspended in 1 ml ice-cold 0.1% NP40 in PBS. Samples were centrifuged for 10 s at 13,000 rpm at 4 °C, supernatant was discarded and the nuclear pellet and the earlier collected whole-cell lysates were prepared for western blot analysis. The whole-cell lysate and nuclear pellets were prepared in sample buffer with β-mE and boiled at 100 °C for 10 min on a thermomixer, shaking at 200 rpm.

**ROS measurement**. Hydrogen peroxide formation was detected using non-fluorescent Dihydrorhodamine 123 (DHR, Invitrogen), which after its oxidation by $H_2O_2$ is converted to fluorescent rhodamine. MEFs were seeded in a 12-well plate, the next day they were treated with PR-619 in the presence or absence of NAC. Cells were then washed with PBS, followed by trypsinisation. Cells were transferred to a 15 ml tube, centrifuged for 3 min at 1600 rpm. The pellet was resuspended in 500 μl DMEM containing the DHR dye (5 μM). Following 30 min incubation at 37 °C, cells were centrifuged again for 3 min at 1600 rpm and the pellet was

resuspended in 2 ml DMEM. Cells were immediately analysed by fluorescence-activated flow cytometry (FACS, Partec). Mean fluorescence values were obtained.

**Mouse tissue**. Ethical approval for mice was granted by the LERC Newcastle University, UK; the work was licensed by the UK Home Office (PPL 60/3864) and complied with the guiding principles for the care and use of laboratory animals. Male, 3 and 24 months old mice of C57Bl6 strain were used in the study. Mice were housed in same-sex cages in groups of 4 to 6 (56 × 38 × 18 cm, North Kent Plastics, Kent, UK) and individually identified by an ear notch. Mice were housed at 20 ± 2 °C under a 12 h light/12 h dark photoperiod with lights on at 7.00 am. The diet used was standard rodent pelleted chow (CRM (P); Special Diets Services, Witham, UK). Paraffin sections were deparaffinised with Histo-Clear (National Diagnostics) and ethanol, antigen was retrieved by incubation in 0.01 M pH 6.0 citrate buffer at 95 °C for 20 min. Slides were incubated in 0.9% $H_2O_2$ for 30 min and afterwards placed in blocking buffer (normal goat serum, Vector Lab) for 30 min at room temperature. Primary antibody (guinea pig α-p62) was applied overnight at 4 °C. Slides were washed three times with PBS and incubated for 30 min with secondary antibody (Vector Lab). Antibodies were detected using peroxidase VECTASTAIN ABC kit (Vector Lab) according to the manufacturer's instructions. Substrate was developed using NovaRED (Vector Lab). Sections were counterstained with haematoxylin and mounted with DPX (Thermo).

**Generation of 'humanised' _D. melanogaster_ Ref(2)P**. CRISPR/Cas9-Mediated Genome Editing (homology-dependent repair (HDR)) using one guide RNA and a dsDNA plasmid donor) was utilised to generate a humanised Ref(2)P (_CG10360_) (WellGenetics, Taiwan). Amino acids 91–116 of Drosophila (knock-in strain $w^{1118}$) Ref(2)P were replaced with amino acids 100–118 of human p62/SQSTM1 peptide. Guide RNA primers: sense 5′-CTTCGACAGAAGGAGCTTCAGCGT-3′; antisense 5′-AAACACGCTGAAGCTCCTTCTGTC-3′.

**Fly husbandry**. Flies were crossed and maintained on standard media at 25 °C in a controlled 12 h light: dark cycle. Male flies were used in all experiments. Survival graphs were created using Graphpad Prism software and log-rank statistics applied. Overall, 60 flies were used for each experimental group and three replicate experiments were performed. Graphs presented depict pooled results.

**Paraquat survival**. Two day old males (20 flies per vial) were starved overnight on 5 ml 1% agar at 25 °C. Flies were transferred to vials containing filter paper soaked in 150 μl of a 5% Sucrose solution with 20 mM Paraquat and maintained at 25 °C. Dead flies were counted at 12 h intervals and media changed daily.

**32 °C survival**. 2 day old males (20 flies per vial) were transferred to 32 °C. Media was changed every 1–2 days and dead flies were counted.

**Chloroquine feeding**. Two day old males (20 flies per vial) were transferred to standard media containing 2.5 mg/ml chloroquine. Flies were maintained at 25 °C for 2 days and then transferred to 32 °C for 2 days. Media was changed every 1–2 days and dead flies were counted.

**p62 alignment and in silico calculations**. p62 protein sequences in 14 organisms (Supplementary Table 3) were identified by searching UniProt for the gene names SQSTM or Ref(2)P, then removing sequence fragments and choosing the longest isoform from each organism. Multiple Sequence alignment was carried out using the Muscle server at EBI (http://www.ebi.ac.uk/Tools/msa/muscle/) with default parameters. The resulting alignment was visualised using ALINE[52]. Conservation is indicated by depth of colour from light-to-dark red, with conservation below 30% indicated in white. Secondary structure in the PB1 domain was indicated by using the X-ray crystal structure (PDB entry 4MJS chain B), and in the ZZ zinc-finger domain using homology to CBP (PDB entry 1TOT).

**Modelling of p62**. We carried out modelling of the p62 construct comprising of PB1 and zinc-finger domains (residues 3–183). Prediction of an initial 3D structure of zinc-finger domain of p62 (residues 121–183) was performed using MOD-ELLER[53] using the crystal structure of human MZM-REP domain of Mib E3 ubiquitin ligase (PDB code: 4XI6) as a template. The sequence identity between the template and the target was 38%. The model has been refined by molecular-mechanical energy minimisation (50,000 cycles) and 10 ns of unrestrained all-atom MD simulation using Gromacs 5.1.2[54] with Amber99SB-ILDN* force field[55] and explicit TIP3P water model. The obtained structure has been fitted on the PB1 domain using protein-protein docking with PatchDock[56], and the top five predicted models were refined by 50,000 cycles of MM minimisation and 5 ns of MD simulation. The obtained models were fitted on cryoEM assemblies of p62 PB1 domains (PDB codes 4UF8 and 4UF9), and the final model has been selected using the geometric constraints (relative positions of cysteines, steric clashes overall the model) and energy-minimised with the disulphide bonds introduced.

**Modelling of the impact of K102E mutation**. The mutation has been introduced to the PB1 domain models (residues 3–125) by manual replacement, followed by the conformational using UCSF Chimera[57]. Then, the whole structure of the mutant (in the monomeric form) has been subjected to 50,000 cycles of MM energy minimisation, followed by 10 ns of all-atom MD in Gromacs 5.1.2 with TIP3P explicit water and the Amber99SB-ILDN* force field[54,55]. The wild-type PB1 domains were subjected to the same protocol, as the control group. In the analysis, we focused on the differences between the interactions in the wild type and the K102E mutant and on the local sampling of the region surrounding the mutated residue.

The modelling of the oligomers was done in UCSF Chimera. The positions of the 102–125 loops were refined using FlexPepDock Monte Carlo approach[58]. The figure has been generated using UCSF Chimera[57].

**Quantification and statistical analysis**. Intracellular aggregates of p62 were scored by eye; the scorer was blinded to the cell treatment and the specific p62 cell line. More than 200 cells were counted per slide and quantification was based on at least three independent experiments unless otherwise stated. For autophagosome (LC3-positive) quantification, z-stack images were collected. Using ImageJ software (NIH), a constant threshold was applied to all the images in the z-stack, and for every image within each experiment. Cells were scored for % cells with more than 20 autophagosomes per cell. At least 30 cells were quantified per condition. For autophagosome quantification by EM, the number of autophagosomes per field of view (images taken at ×25k magnification) was quantified by eye from 20 images.

Quantification of immunoblots was carried out using ImageJ software (NIH). Two-tailed, unpaired Student's t-tests were carried out on experimental data from at least three individual experiments. Fly survival curves were made using GraphPad Prism and log-rank tests were used for statistical analysis.

**Data availability**. The authors declare that all the other data supporting the findings of this study are available within the article and its supplementary information files and from the corresponding author upon request.

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

## Acknowledgements

We are grateful to G. Nelson, E. Bennett, S. Drain, L. Xie, M. Johnson, S. Keers, M. Monteiro, T. Alder, R. Hall and L. Ramsay for technical assistance; Y. Rabanal Ruiz for help with cell culture and imaging; R. Layfield, T. Johansen, G. Nelson and Addgene for plasmids; I. Bjedov and I. Nezis for antibodies. This work was funded by BBSRC (V.I.K., A.S. and J.F.P), the NIHR Newcastle Biomedical Research Centre, the Newcastle upon Tyne Hospitals NHS Charity (V.I.K.) and ERC (A.S). D.C.R is grateful for funding from the UK Dementia Research Institute (funded by the MRC, Alzheimer's Research UK and the Alzheimer's Society). D.J. is funded by a Newcastle University Faculty of Medical Sciences Fellowship.

## Author contributions

B.C., E.G.O., D.M. and R.S. performed the majority of experiments. F.M.M., D.J., N.K. and J.F.P. performed and evaluated individual experiments; S.W., J.F.P., J.A., E.A.V., E.T., D.S. and S.M. provided materials; G.R.S and A.K.B. generated alignments and the computational model. E-L.E. analysed EM data. V.I.K, A.S., D.C.R. and E.A.V designed and supervised the study; V.I.K. supervised the entire project and wrote the paper with help from all authors.

## Additional information

**Competing interests:** The authors declare no competing financial interests.

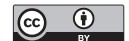

