## [Peer Review File · Nature Communications]

Reviewer #1 (Remarks to the Author):

This manuscript describes the novel discovery that vertebrate p62, an autophagic receptor molecule, uses oxidative stress mediated disulfide bond formation to stimulate pro-survival autophagy pathways, thereby increasing oxidative stress resistance and stress survival. The study is a true tour de force, providing detailed insights into the cysteine residues responsible for redox sensing in p62, revealing the downstream effects of the oxidation on autophagy and protein degradation, and demonstrating that introduction of the critical cysteines into *Drosophila* p62 mediates vertebrate p62-like redox sensitivity, and with that increases oxidative stress-induced autophagy and stress resistance in flies. Lastly, the authors provide some evidence that perturbation of the redox sensing mechanism in p62 might contribute to age-related pathologies observed in SALS patients. This is an impressive piece of work, and I have only a few minor comments that the authors might want to address:

1) There is some amount of inconsistency in the results when it comes to the degree of DLC formation; this is probably best illustrated in Fig. S1A/1C where exposure of HeLa cells to 10 μ M PR-619 causes massively different amounts of DLC formation; another example is Fig. 3b (lane 1-4), which shows no DLC formation in response to 3 mM H₂O₂ for 1 to 30 min whereas Fig. 1C shows significant DLC formation in response to 500 μ M H₂O₂ for 10 min. While it is clear that DLC form in response to oxidant, the inconsistencies in the results are somewhat distracting. Also, why use HEK cells in some studies and HeLa cells in others?

2) The authors might want to comment on the working mechanism of PR-619. While I can see how treatment of cells with PR-619 might disturb redox homeostasis and increase oxidative stress in vivo, it is unclear to me how, based on the chemistry of PR-619, it directly oxidizes purified p62 in vitro (Fig. S1E). The authors might also want to consider expressing p62 with a His-tag rather than a GST fusion, which might respond to PR-619 as well.

In this regard, it would be interesting to see whether thioredoxin inhibitors equally delay the reduction of the DLC formed in response to PR-619.

3) Some of the figures/images require more explanation or improved color codes

Fig. 1B: It is not clear where the aggregates are supposed to be located, which parts of the images were quantified and what the arrow heads are pointing to.

Fig. 1D: the statement that "formation of p62 DLC in response to H₂O₂ or PR-619 treatment correlates with accumulation of intracellular p62 aggregates" is not backed up by the data shown.

Fig. 3D: are these images taken in response to peroxide treatment?

Fig. 3E; blue and green is very difficult to distinguish in print (or on the computer).

Fig. S9: is this analysis been done in response to peroxide treatment? Also, are similar results obtained in response to PR-619 treatment?

Reviewer #2 (Remarks to the Author):

In this manuscript, the authors found that p62 causes the formation of disulphide-linked conjugates (DLC) in response to oxidative stress. This disulfide bonds were formed by C105 and C113, highly conserved cysteine residues among vertebrate homologs of p62. Double mutant of p62 C105A, C113A expressing cells are highly susceptible to oxidative stress, while flies expressing p62 containing C105 and C113 exhibits the highest oxidative resistance. Importantly, mutation of K102E that has been identified as causative in sporadic ALS (SALS), impaired the DLC formation of p62 under oxidative stress condition, potentially representing a pathological mechanism in SALS. On the basis of these observations, the authors conclude that p62 senses the generation of reactive oxygen species (ROS), and increase autophagy for cell protection against oxidative stress. Besides the potential interest of the results, the presented data are not sufficient to support the author's model and lacks several important analyses. Therefore, I cannot support publication of this study in Nature Communications, at least current version.

1. The molecular mechanisms of the resistance to ROS by disulphide-linked p62 remain entirely unknown. The authors should show the activity of p62-Keap1-Nrf2 pathway by Western blot analysis with anti-phospho-p62 antibody or qPCR analysis of Nrf2 target genes in Fig.3B (Ref. Mol Cell.51:618-631.2013).

2. C105 and C113 residues are not essential for the formation of p62 aggregates depending on the PB1 domain. The authors should characterize and explain the distinct roles of oligomerized p62 DLC-dependent and PB1-dependent.

3. Expression of p62 could be up-regulated by Keap1-Nrf2 pathway under oxidative stress conditions (Ref. J Biol Chem.285:22576-22591.2010). Increased p62 may be associated with the

formation of p62 DLC and increased autophagy in old mouse. The authors should examine the expression levels of p62 in old and young mouse.

4. The authors should determine the 3D structures of 1-122aa region of p62 or which including K102E mutation.
5. The action mechanism related to cysteine oxidation by PR619 is unknown. The authors should conduct the interaction assay of p62 and PR619 by using biotinylated PR619.
6. p62 aggregates via PB1 can be easily detected by microscopy. Why do the authors not detect the aggregates formed by disulfide bonds of p62C105C113.
7. Experiments of Fig.S1E require recombinant of p62C105AC113A mutant as a negative control.
8. In Fig.S1 and S4B, the authors should indicate the results obtained in the reduced conditions.
9. In Fig.3C, the authors should make significant differences of ubiquitin or LC3 between normal and oxidative stress conditions, like the results of p62.
10. Page9, line-18, there is a typo-error; survival → survival.

Reviewer #3 (Remarks to the Author):

The manuscript by Manni et al. contains important novel insights on the interplay between autophagy and oxidative stress, and as such deals with an important topic which is worth consideration in this journal.

The experimental evidence supporting the interpretation is extensive, however some important details do not fit with their model and these inconsistencies should be addressed prior to publication.

The core concept of this manuscript is that p62 DLC formation is priming p62 aggregation. However this is not consistent with some of their data. In Figure 1D the p62 puncta are not quantified differently from most figures. The authors should quantify this critical information.

However, if these images are representative, H2O2 induces many more puncta than PR-619. That is the opposite of what is observed in Fig 1C, in which PR-619 induces a greater formation of DLC. If the two phenomena do not correlate, the authors cannot rule out that they are not independent of each other, which is in contrast with their analysis of the p62 C105,113A. As this inconsistency lies at the basis of the whole investigation, it has to be addressed and solved to validate the model. The authors have to make it clear whether formation of DLC and of visible aggregates correlate or not.

In Figure 3C the most important comparisons are those between the two mutant p62 and the wt p62. For instance in LC3-II/Actin under H2O2 is the difference between wt p62 and any of the two mutants statistically significant? If not, the authors cannot claim that the two mutants significantly differ from WT p62 in their ability to restore autophagy. In general in all the Fig 3 graphs and beyond the authors should compare the mutants to WT p62, not to the p62 $-/-$, because this is the comparison that makes sense in this context.

The analysis of autophagy flux in *Drosophila* is not adequately developed and the little that is shown is not entirely supportive. In Fig S10B CQ does not cause an increase in Atg8, in the WT. This suggest that CQ has not worked in this condition. As such, it is not possible to claim any different effect under the Ref(2)Pox because the baseline for comparison is abnormal. The authors should explain this and provide a more detailed analysis of the autophagy flux to match that in He-La cells.

In Fig S12 there is a number of issues. Panel D is not quantified and it should, because from those pictures I draw the opposite conclusion from that of the authors. It appears that the K102E mutants has the same, if not more puncta for LC3 and p62. Also in the human material in E, the p62 monomers have a molecular weight between 37 and 50 kDa. This suggests massive protein degradation. As such the presence of lower MW species as DLC is likely an artefact of tissue preservation rather than an effect of the K102E mutation on DLC formation and would be against the proposed explanation for ALS.

Reviewer #1 (Remarks to the Author):

This manuscript describes the novel discovery that vertebrate p62, an autophagic receptor molecule, uses oxidative stress mediated disulfide bond formation to stimulate pro-survival autophagy pathways, thereby increasing oxidative stress resistance and stress survival. The study is a true tour de force, providing detailed insights into the cysteine residues responsible for redox sensing in p62, revealing the downstream effects of the oxidation on autophagy and protein degradation, and demonstrating that introduction of the critical cysteines into *Drosophila* p62 mediates vertebrate p62-like redox sensitivity, and with that increases oxidative stress-induced autophagy and stress resistance in flies. Lastly, the authors provide some evidence that perturbation of the redox sensing mechanism in p62 might contribute to age-related pathologies observed in SALS patients. This is an impressive piece of work, and I have only a few minor comments that the authors might want to address:

1) There is some amount of inconsistency in the results when it comes to the degree of DLC formation; this is probably best illustrated in Fig. S1A/S1C where exposure of HeLa cells to 10 μ M PR-619 causes massively different amounts of DLC formation; another example is Fig. 3b (lane 1-4), which shows no DLC formation in response to 3 mM H₂O₂ for 1 to 30 min whereas Fig. 1C shows significant DLC formation in response to 500 μ M H₂O₂ for 10 min. While it is clear that DLC form in response to oxidant, the inconsistencies in the results are somewhat distracting. Also, why use HEK cells in some studies and HeLa cells in others?

We agree with the reviewer that the extent of DLC formation is variable. We believe that this is a result of the dynamic process of protein oxidation and the relatively unstable nature of oxidising agents, particularly hydrogen peroxide which decays over time. We have attempted to minimise this variability by only ever using a stock H₂O₂ for a maximum of two weeks which, whilst minimizing this variability, did not completely extinguish it. We have also corrected a mistake noted in the legend of Figure 1C, where 3mM rather than 500 μ M H₂O₂ was used, leading to confusion.

We initially used HeLa and HEK293E cells to demonstrate that similar effects occurred in different cell lines. However, we agree with the reviewer that in the interests of consistency the results in a single cell line should be presented. We have now replaced the blot in Figure S3A with an H₂O₂ time course in HeLa rather than HEK293E cells. Please note the results are in line with those seen in HEK293E indicating that H₂O₂ causes a rapid, concentration-dependent induction of p62 DLC within 20 seconds. These resolve over time at lower doses of H₂O₂ but persist at higher levels of oxidative stress potentially reflecting saturation of cellular antioxidant systems.

2a) The authors might want to comment on the working mechanism of PR-619. While I can see how treatment of cells with PR-619 might disturb redox homeostasis and increase oxidative stress in vivo, it is unclear to me how, based on the chemistry of PR-619, it directly oxidizes purified p62 in vitro (Fig. S1E).

We apologise for not clarifying this in the original manuscript but the compound is a redox cyclor (see <http://www.sciencedirect.com/science/article/pii/S0003269715001189>) and as

such catalyzes the oxidation of protein thiols in aqueous solutions. It acts as an electron shuttle through reduction to dihydropyridine and reoxidation. This has now been clarified in the text.

2b) The authors might also want to consider expressing p62 with a His-tag rather than a GST fusion, which might respond to PR-619 as well.

We attempted to express the His-tagged version of p62 *in vitro* but the protein was insoluble. It is likely that the larger GST tag assists in maintaining p62 in the soluble form. Isolation of the His-tagged construct will require further optimisation which was not possible within the timeframe of the revisions. However, we have demonstrated oxidation of recombinant GST-p62 *in vitro*, as well oxidation of Flag-tagged, GFP-tagged and untagged endogenous protein in cells throughout the manuscript. We have also produced a cysteine mutant GST-tagged p62 which shows defective oxidation compared to wild type protein *in vitro* (new Figure S5D). We are therefore confident that the type of tag (or the absence thereof) is not affecting the oxidation process.

2c) In this regard, it would be interesting to see whether thioredoxin inhibitors equally delay the re-reduction of the DLC formed in response to PR-619.

PR-619 in cell culture, unlike H₂O₂, induces persistent oxidative stress with no detectable reduction of DLC over time (e.g. Figure S3C). Instead, the opposite is seen, where gradually most of the protein becomes oxidised. Therefore there is no evidence that thioredoxin inhibitors would affect DLC.

3) Some of the figures/images require more explanation or improved color codes Fig. 1B: It is not clear where the aggregates are supposed to be located, which parts of the images were quantified and what the arrow heads are pointing to.

We thank the Reviewer for highlighting this. We have now replaced the original images with higher resolution images and improved the labelling.

Fig. 1D: the statement that “formation of D62 DLC in response to H₂O₂ or PR-619 treatment correlates with accumulation of intracellular p62 aggregates” is not backed up by the data shown.

We apologise if this statement is misleading. Indeed, our data does not suggest direct correlation of DLC formation and aggregate formation, as the latter process also requires the function of the PB1 domain, the absence of which prevents aggregation in response to any stimuli (Figure S8). At this stage of the manuscript (i.e. Figure 1), the data only suggests that DLC may contribute to aggregate formation. We have now replaced the original statement with: “In conditions of p62 DLC formation, we also observed an increase in intracellular p62 aggregates compared to untreated controls”. To support this statement, we have included quantification of p62 aggregates in Figure 1D which shows that upon the treatment with pro-oxidants, H₂O₂ and PR-619 approximately 25-30% cells contain p62 aggregates as compared to 15% in control conditions. To highlight p62 aggregates, we have included zoomed in images in all conditions.

Fig. 3D: are these images taken in response to peroxide treatment?

The images in Figure 3E (previously Figure 3D) were taken in control conditions which has now been clarified in the figure legend. We have added quantification of autophagosomes in all cell lines as requested by Reviewer 3 (Figure 3F). As you can see the % of cells with >20 autophagosomes/cell correlates with Western blot data in Figure 3C and 3D which shows the increase in levels of LC3-II, supporting our conclusion that wild-type p62 (but not oligomerisation-deficient C105/113A and K7A/D69A mutants) promotes autophagy.

Fig. 3E; blue and green is very difficult to distinguish in print (or on the computer).

We have modified the representative images and 'dead' cells are now shown in yellow while all nuclei are in blue. We also show the channels separately so that the differences in 'dead' cell number can be more easily appreciated.

Fig. S9: is this analysis been done in response to peroxide treatment? Also, are similar results obtained in response to PR-619 treatment?

Degradation assays (Figure 3A and S9A) were carried out following treatment of cells with H_2O_2 which has now been made clearer within the text. We have confirmed the delayed degradation kinetics of the C105A,C113A mutant compared to wild type p62 in the presence of PR-619 (new Figure S9B). We have included additional data to support the conclusions from Figures 3 and 4, that p62 DLC is required for efficient autophagy. Thus, our new data indicate that blocking autophagy using bafilomycin A1 or chloroquine cancels the difference between the rates of wild type and mutant p62 degradation thus confirming that faster degradation of wild type p62 is mediated by autophagy (new Figures 3B and S9C).

Reviewer #2 (Remarks to the Author):

In this manuscript, the authors found that p62 causes the formation of disulphide-linked conjugates (DLC) in response to oxidative stress. This disulfide bonds were formed by C105 and C113, highly conserved cysteine residues among vertebrate homologs of p62. Double mutant of p62 C105A, C113A expressing cells are highly susceptible to oxidative stress, while flies expressing p62 containing C105 and C113 exhibits the highest oxidative resistance. Importantly, mutation of K102E that has been identified as causative in sporadic ALS (SALS), impaired the DLC formation of p62 under oxidative stress condition, potentially representing a pathological mechanism in SALS. On the basis of these observations, the authors conclude that p62 senses the generation of reactive oxygen species (ROS), and increase autophagy for cell protection against oxidative stress. Besides the potential interest of the results, the presented data are not sufficient to support the author's model and lacks several important analyses. Therefore, I cannot support publication of this study in Nature Communications, at least current version.

1. The molecular mechanisms of the resistance to ROS by disulphide-linked p62 remain entirely unknown. The authors should show the activity of p62-Keap1-Nrf2 pathway by Western blot analysis with anti-phospho-p62 antibody or qPCR analysis of Nrf2 target genes in Fig.3B (Ref. Mol Cell.51:618-631.2013).

We thank the Reviewer for this important comment. As requested, we have investigated the Nrf2 response to H₂O₂ treatment in our cell lines and could not see a correlation with increased cell survival of cells expressing wild type vs mutant p62. As can be appreciated in Figure for Reviewer (Fig. 1) included in this document, H₂O₂ treatment leads to robust upregulation of the Nrf2 pathway indicated by increases in total and nuclear Nrf2 (Fig. 1A, B), as well as induction of Nrf2 target gene *HO-1* (Fig. 1C). However, unlike in the study cited by the reviewer (Mol Cell.51:618-631.2013) where Nrf2 was shown to be induced in p62-dependent manner in response to As(III), we do not observe a significant p62 dependence in response to H₂O₂ in our cells. Most importantly, we do not see significant difference in Nrf2 response between cells expressing wild type and the Cys mutant of p62. Additionally, treatment of cells with retinoic acid, an inhibitor of Nrf2 signalling (Wang *et al.*, 2007), did not negate the rescue of cell death in response to H₂O₂ in cells expressing wild type p62 (Fig. 1D). Collectively, these data do not support that Nrf2 signalling is underlying this mechanism of stress resistance by p62.

Fig. 1
Fig. 1. (A) *p62*^{-/-} MEFs stably expressing FLAG-tagged wild type, C105A,C113A or K7A,D69A PB1 domain mutant p62 were treated with H₂O₂ (1mM) in serum free media for 5 hours and subjected to a nuclear fractionation followed by immunoblot analysis for Nrf2, Histone 3 and GAPDH and quantified (B). (C) Cells were treated as in (A) and Nrf2 target gene *HO-1* mRNA levels were analysed by qPCR, actin was used as a loading control. (D) *p62*^{-/-} MEFs stably expressing FLAG-tagged wild type or C105A,C113A mutant p62 were pre-treated with retinoic acid (RA, 30μM) for one hour in serum free media followed by the same treatment as in (A) with or without retinoic acid (30μM) and % cell death was analysed by ReadyProbes fluorescent dyes (Life Technologies).

In a parallel set of experiments we investigated if, as originally proposed, upregulation of autophagy acts as the mechanism of resistance to oxidative stress. Inhibition of autophagy using chloroquine or bafilomycin A1 completely cancelled out the difference in the rates of cell death between wild type and C105A,C113A p62 (new Figures 3B and S9C) as well as the cell survival in response to H₂O₂ treatment between wild type and mutant p62 (new Figure 3I). Thus, we believe that our data strongly implicate the upregulation of autophagy by disulphide-linked p62 as a mechanism of increased cell survival due removal of misfolded proteins and reduction of oxidative stress. This conclusion fits well with published studies which demonstrate that turnover of misfolded proteins via autophagy or through the proteasome is sufficient to reduce oxidative stress and promote cell survival (e.g. Cell Rep, 18, 13, p3143–3154, 2017).

2. C105 and C113 residues are not essential for the formation of p62 aggregates depending on the PB1 domain. The authors should characterize and explain the distinct roles of oligomerized p62 DLC-dependent and PB1-dependent.

Indeed, our data indicates that p62 aggregation dependent on oligomerisation of the PB1 domain does not require p62 oxidation and DLC formation. In contrast, PB1 domain interactions are required for the formation of p62 aggregates in response to oxidative stress (Figure S8A). This suggests a sequential order of events where oxidation of p62 induces an initial oligomerisation event leading to PB1 domain-dependent formation of aggregates.

Importantly, cells expressing oxidation-insensitive or PB1 domain defective mutants of p62 appear to have similar functional phenotypes (e.g. autophagy and cell survival defects, Figure 3) and therefore although triggered by different stimuli p62 oxidation and PB1 dependent oligomerisation serve to increase cell survival. This suggests that, irrespective of the underlying molecular mechanism, oligomerisation of p62 is required for its function. In this case, oxidation of p62 is one of the mechanisms to oligomerise the protein which acts specifically in conditions of oxidative stress as a trigger for this event. We have modified the discussion to make this point clearer.

3. Expression of p62 could be up-regulated by Keap1-Nrf2 pathway under oxidative stress conditions (Ref. J Biol Chem.285:22576-22591.2010). Increased p62 may be associated with the formation of p62 DLC and increased autophagy in old mouse. The authors should examine the expression levels of p62 in old and young mouse.

We agree with the Reviewer that the phenotypes in mouse brains that we observed may indicate increased autophagy. However, we have performed qPCR on brain tissue from young and old mice as suggested by the Reviewer (new Figure S1A) and did not observe a significant difference in p62 expression. It is still possible that increased DLC formation reflects changes in autophagy as well as increased oxidative stress in aged mice, however this would require additional investigations *in vivo*, which we believe are outside the scope of this study.

4. The authors should determine the 3D structures of 1-122aa region of p62 or which including K102E mutation.

We thank the Reviewer for this excellent suggestion which allowed us to propose a potential model of p62 DLC and refine our hypothesis regarding pathology due to the K102E mutation. We have used the existing 3D structure of N-terminal region of p62 (Ciuffa et al, Cell Reports, 2015) to map the localisation of oxidation-sensitive Cys residues and the K102E mutation (new Figure S12A). The 20aa C-terminal segment is unstructured (as in our prediction, Figure S5A) however the position of this segment within the model allows for intermolecular interactions between Cys 105 and 113 residues contained within this region. Moreover, this model proposes a mechanism by which K102E may affect disulphide-mediated oligomerisation of p62 (please see the text).

5. The action mechanism related to cysteine oxidation by PR619 is unknown. The authors should conduct the interaction assay of p62 and PR619 by using biotinylated PR619.

PR619 is a strong oxidizing agent which produces ROS in aqueous solutions (please see <http://www.sciencedirect.com/science/article/pii/S0003269715001189> and our response to a similar comment from Reviewer 1). We apologise for not making this clear previously and this explanation has now been added to the text. The mechanism of thiol oxidation does not involve direct binding to the target protein and therefore we did not perform interactions assays as proposed by the Reviewer.

6. p62 aggregates via PB1 can be easily detected by microscopy. Why do the authors not detect the aggregates formed by disulfide bonds of p62C105C113.

Similar comments were also made by the other Reviewers and we agree that we may not have made our interpretation of the data clear in the text and this has now been improved. We do not imply that oligomerisation of p62 via DLC equates to the formation of detectable aggregates. DLC are relatively small order oligomers, at least at low levels of oxidative stress, and would not be detected as aggregates using microscopy. However, oxidation of p62 can potentially seed and promote the formation aggregates detectable by microscopy (e.g. Figure 2C), a process requiring higher order oligomer assembly mediated by the PB1 domain-dependent non-covalent interactions (Figure S8A).

7. Experiments of Fig.S1E require recombinant of p62C105AC113A mutant as a negative control.

We thank the Reviewer for his comment and have now conducted these experiments. As you can see in new Figure S5D, the C105A,C113A mutant has a reduced capacity to form DLC in response to oxidation by H₂O₂ or PR-619 *in vitro*.

8. In Fig.S1 and S4B, the authors should indicate the results obtained in the reduced conditions.

As requested by the Reviewer we have now included data produced in reduced conditions (new Figures S1B, D, E and F and S4B).

9. In Fig.3C, the authors should make significant differences of ubiquitin or LC3 between normal and oxidative stress conditions, like the results of p62.

We now show the statistical difference between treatment groups as requested by the Reviewer. Please also note that, as suggested by Reviewer 3, we have changed the statistical comparison to wild type p62, not to the p62^{-/-} null cells.

10. Page9, line-18, there is a typo-error; survival → survival.

We apologise for this mistake and have now corrected this.

Reviewer #3 (Remarks to the Author):

The manuscript by Manni et al. contains important novel insights on the interplay between autophagy and oxidative stress, and as such deals with an important topic which is worth consideration in this journal.

The experimental evidence supporting the interpretation is extensive, however some important details do not fit with their model and these inconsistencies should be addressed prior to publication.

The core concept of this manuscript is that p62 DLC formation is priming p62 aggregation. However this is not consistent with some of their data. In Figure 1D the p62 puncta are not quantified differently from most figures. The authors should quantify this critical information. However, if these images are representative, H₂O₂ induces many more puncta than PR-619. That is the opposite of what is observed in Fig 1C, in which PR-619 induces a greater formation of DLC. If the two phenomena do not correlate, the authors cannot rule out that they are not independent of each other, which is in contrast with their analysis of the p62 C105,113A. As this inconsistency lies at the basis of the whole investigation, it has to be addressed and solved to validate the model. The authors have to make it clear whether formation of DLC and of visible aggregates correlate or not.

We agree with the Reviewer that there was an apparent inconsistency. We have now repeated and quantified p62 aggregation in response to different treatments which is presented in new Figure 1D. As can be seen from these data, PR-619 is a stronger inducer of p62 aggregate formation than H₂O₂, this correlates with the effect of these treatments on DLC formation. As outlined in our response to Reviewer 1 point 3 regarding Figure 1D and Reviewer 2 point 6, we do not assume a direct correlation between DLC formation and p62 aggregation as detected by microscopy. The former process, at least initially results in the formation of low order oligomers which would not be detectable as aggregates by microscopy. At the same time it could stimulate/seed p62 aggregates, the latter process also requiring PB1 domain-mediated non-covalent interactions.

In Figure 3C the most important comparisons are those between the two mutant p62 and the wt p62. For instance in LC3-II/Actin under H₂O₂ is the difference between wt p62 and any of the two mutants statistically significant? If not, the authors cannot claim that the two mutants significantly differ from WT p62 in their ability to restore autophagy. In general in all the Figure 3 graphs and beyond the authors should compare the mutants to WT p62, not to the p62^{-/-}, because this is the comparison that makes sense in this context.

We agree with the Reviewer that comparison to wild type p62 is more meaningful. We have performed additional experiments for Figure 3 and all statistical analyses have been modified in accordance with the Reviewer's suggestion.

The analysis of autophagy flux in Drosophila is not adequately developed and the little that is shown is not entirely supportive. In Fig S10B CQ does not cause an increase in Atg8, in the WT. This suggest that CQ has not worked in this condition. As such, it is not possible to

claim any different effect under the Ref(2)Pox because the baseline for comparison is abnormal. The authors should explain this and provide a more detailed analysis of the autophagy flux to match that in He-La cells.

We thank the Reviewer for highlighting this discrepancy. We have now repeated CQ experiments at 32°C and can detect an increase in Atg8-II levels in all genetic backgrounds in line with the previously shown result, whilst the increase in Ref(2)Pox flies is significantly stronger (new Figure S10B), supporting our original conclusion that autophagy flux is increased in Ref(2)Pox flies. We have also added the Atg8 blots for all the fly data (PQ, 32°C and 7 vs 20 day old). In accordance with our original conclusion there is no significant difference in Atg8-II between wild type and Ref(2)Pox flies in the absence of autophagy blocker. Together, these data suggest an increased Atg8-II turnover in Ref(2)Pox flies. The levels of Ref(2)P and ubiquitin in our fly model, which were included in our original submission, are in line with analyses in HeLa cells.

In Fig S12 there is a number of issues. Panel D is not quantified and it should, because from those pictures I draw the opposite conclusion from that of the authors. It appears that the K102E mutants has the same, if not more puncta for LC3 and p62.

We have replaced the images with higher quality confocal images which show less non-specific staining of LC3 and p62 which previously positively stained puncta in the nucleus of K102E mutant cells unlike the cytoplasmic autophagosomal vesicles in wild type p62 cells. Therefore by definition these puncta do not correspond to autophagosomes. Please see example images in new Figure S12D. As requested, we have also quantified the number of autophagosomes. Furthermore, to be consistent, we have quantified the number of autophagosomes in other p62 cell lines (Figure 3E, F).

Also in the human material in E, the p62 monomers have a molecular weight between 37 and 50 kDa. This suggests massive protein degradation. As such the presence of lower MW species as DLC is likely an artefact of tissue preservation rather than an effect of the K102E mutation on DLC formation and would be against the proposed explanation for ALS.

We agree with the Reviewer that the reduced molecular weight of monomeric p62 in human spinal cord may result from degradation in post-mortem tissue although it could also result from an alternative splicing event (Dr. L.J. Hocking, personal communication). However, bands of a similar size, and in some instances (e.g. SALS1) even lower molecular weight bands, are detected in control and SALS tissues where no significant band of ~150 kDa is observed. This argues against the band of ~150 kDa being a result of protein degradation. We believe that this data, despite its limitations, suggests a potential under oxidation of p62 in the K102E case.

Reference:

Wang, X.J., Hayes, J.D., Henderson, C.J. and Wolf, C.R. (2007) 'Identification of retinoic acid as an inhibitor of transcription factor Nrf2 through activation of retinoic acid receptor alpha', *Proc Natl Acad Sci U S A*, 104(49), pp. 19589-94.

Reviewer #1 (Remarks to the Author):

The authors have adequately addressed my comments.

Reviewer #3 (Remarks to the Author):

As I have mentioned in my previous review, Manni et al provide important novel insights on the intersection between oxidative stress and autophagy.

In their revised manuscript they have satisfactorily addressed all my concerns, except one, the last. I am still of the same opinion that the data provided in Figure S12E cannot be used as a confirmation of their pathological model for the K102E mutation in ALS.

Unfortunately, too many reasonable doubts remain over their statements regarding this figure, which do not warrant acceptability in this journal.

It remains a speculation that the bands between 50 and 37kDa in this figure are p62 monomers and not fragments. It cannot be accepted on the basis of a personal communication that they may be products of alternative splicing. I stress that if this is the case, I do not see any band at 62kDa, which should be the real p62 monomer. Are the authors saying that in human spinal cord p62 is actually all alternatively spliced so that a full length p62 never forms? This is a big claim and should be proved for publication.

Regarding the 150 KDa band, there is also some in other SALS patients without the K102E mutation. In the K102E sample it is rather a greater smear, which is more in agreement with protein degradation than with a neat trimeric format.

Anyhow, the full scale of the issue is that, with just these data, the authors cannot rule out that what they observe are merely different degrees of protein degradation in problematic human postmortem samples.

In consideration of this, if they want to conclude that they have human data that support their pathological model for K102E they would need to provide evidence that what they call monomers are indeed monomers, and not fragments, and that the 150KDa smear is indeed a trimer of those monomers.

However, it would be unfair to make this a requirement for publication. I believe this is a vast and solid paper that has a number of exceptionally interesting findings and also provides an intellectually stimulating model for the K102E mutation. It just falls short of having human data to support this model. I suggest the authors can go as far as to propose their model for K102E, but then leave the confirmation of this model on more solid ground as a future prospect and significantly rewrite this part.

Reviewer #4 (Remarks to the Author):

In this manuscript, the authors found that p62 makes intermolecular disulfide bonds in response to oxidative stress, and then induces autophagy to remove p62 protein aggregates, conferring resistance to oxidative stress. This novel mechanism of disulfide-linked p62 against oxidative stress is highly conserved in mammals but not in fly. The authors demonstrated that disulfide-linked p62 increases autophagy and improves the susceptibility to stress in flies. Moreover, the authors revealed that p62 mutant lacking disulfide bonds is involved in the pathology of SALS. On the basis of these results, the authors propose that oxidized p62 facilitates the intracellular

clearance through p62-selective autophagy. There remain some issues to be solved in this study.

(1) The authors claim that disulfide-linked p62 aggregate is a critical factor in autophagy induction under the oxidative conditions. However, it is apparent that p62 forms self-oligomer through the interaction of PB1 domain as shown in structural and EM analyses (Ciuffa R., et al. Cell Rep.11, 748–758. 2015, Michael W., et al. Mol Cell. 12, 39-50. 2003). Therefore, the disulfide bonds of p62 could be formed by oxidization of oligomerized p62. Indeed, normal PB1 is required for the formation of disulfide-linkage of p62 as shown in Fig 3 E. If the authors claim that p62 facilitates their disulfide-linkage under the oxidative conditions, the oligomerized structure of p62 K7A, D69A mutant should indicate in this study by similar methods in published previously (Ciuffa R., et al. Cell Rep.11, 748–758. 2015).

(2) In Fig.2 and Fig3, I am wondering the affect of C105A, C113A mutations on PB1 oligomer formation. To exclude this possibility, the authors should indicate the oligomer formation of p62 mutant by some experiment such as IP, Native-PAGE, or gel filtration chromatography.

(3) Nrf2 is mainly activated by two distinct pathways, which regulated by p62-dependent and p62-independent system. In normal cells, Nrf2 is usually activated by Keap1-Nrf2 axis respond to oxidative stress. The authors should indicate the regulatory mechanism of Nrf2 activation in the cells producing disulfide-linked p62. It could be easily determined by immunoblotting with anti-phospho p62 (pS349) antibody (Ichimura Y. et al. Mol Cell. 51, 618-631. 2013).

(4) In this study, autophagy is seemed to be increased depending on the accumulation of disulfide-linked p62 aggregates as presented in Fig. 3. However, co-localization of p62 and LC3 signals is not exactly correlated with the activation of autophagy, because p62 directly binds to LC3 (Pankiv S. et al. J Biol Chem. 282, 24131-24145. 2007). For this, the authors should judge the autophagy activity by the increased number of autophagosomes and autolysosomes with EM analysis.

Reviewers' comments:

Reviewer #1 (Remarks to the Author):

The authors have adequately addressed my comments.

Reviewer #3 (Remarks to the Author):

As I have mentioned in my previous review, Manni et al provide important novel insights on the intersection between oxidative stress and autophagy.

In their revised manuscript they have satisfactorily addressed all my concerns, except one, the last. I am still of the same opinion that the data provided in Figure S12E cannot be used as a confirmation of their pathological model for the K102E mutation in ALS.

Unfortunately, too many reasonable doubts remain over their statements regarding this figure, which do not warrant acceptability in this journal.

It remains a speculation that the bands between 50 and 37kDa in this figure are p62 monomers and not fragments. It cannot be accepted on the basis of a personal communication that they may be products of alternative splicing. I stress that if this is the case, I do not see any band at 62kDa, which should be the real p62 monomer. Are the authors saying that in human spinal cord p62 is actually all alternatively spliced so that a full length p62 never forms? This is a big claim and should be proved for publication.

Regarding the 150 KDa band, there is also some in other SALS patients without the K102E mutation. In the K102E sample it is rather a greater smear, which is more in agreement with protein degradation than with a neat trimeric format.

Anyhow, the full scale of the issue is that, with just these data, the authors cannot rule out that what they observe are merely different degrees of protein degradation in problematic human postmortem samples.

In consideration of this, if they want to conclude that they have human data that support their pathological model for K102E they would need to provide evidence that what they call monomers are indeed monomers, and not fragments, and that the 150KDa smear is indeed a trimer of those monomers.

However, it would be unfair to make this a requirement for publication. I believe this is a vast and solid paper that has a number of exceptionally interesting findings and also provides an intellectually stimulating model for the K102E mutation. It just falls short of having human data to support this model. I suggest the authors can go as far as to propose their model for K102E, but then leave the confirmation of this model on more solid ground as a future prospect and significantly rewrite this part.

We have followed the advice from the Reviewer and removed the data using human spinal cord tissue. We have instead suggested a validation of our model in human patients as a future work.

Reviewer #4 (Remarks to the Author):

In this manuscript, the authors found that p62 makes intermolecular disulfide bonds in response to oxidative stress, and then induces autophagy to remove of p62 protein aggregates, conferring resistance to oxidative stress. This novel mechanism of disulfide-linked p62 against oxidative stress is highly conserved in mammals but not in fly. The authors demonstrated that disulfide-linked p62 increases autophagy and improves the susceptibility to stress in flies. Moreover, the authors revealed that p62 mutant lacking disulfide bonds is involved in the pathology of SALS. On the basis of these results, the authors propose that oxidized p62 facilitates the intracellular clearance through p62-selective autophagy. There remain some issues to be solved in this study.

(1) The authors claim that disulfide-linked p62 aggregate is a critical factor in autophagy induction under the oxidative conditions. However, it is apparent that p62 forms self-oligomer through the interaction of PB1 domain as shown in structural and EM analyses (Ciuffa R., et al. Cell Rep.11, 748–758. 2015, Michael W., et al. Mol Cell. 12, 39-50. 2003). Therefore, the disulfide bonds of p62 could be formed by oxidization of oligomerized p62. Indeed, normal PB1 is required for the formation of disulfide-linkage of p62 as shown in Fig 3 E. If the authors claim that p62 facilitates their disulfide-linkage under the oxidative conditions, the oligomerized structure of p62 K7A, D69A mutant should indicate in this study by similar methods in published previously (Ciuffa R., et al. Cell Rep.11, 748–758. 2015).

We believe that this is a misunderstanding. Our data, and based on the structural model described in Supplementary Figure S12A, suggest that the disulphide-dependent oligomerisation of p62 supports different structures to those mediated purely by PB1 domain interactions as shown by Ciuffa et al. Indeed, we show that PB1 domain-mediated interactions are not required for the formation of disulphide linkages of p62 (Supplementary Figure S5C). In the Discussion we proposed that the two processes can act relatively independently, however can contribute to the formation of p62 aggregates. We would therefore request not to undertake these structural studies. Furthermore, since we observe a mix of oligomeric species, of different sizes, the suggested experiment would not be technically possible.

(2) In Fig.2 and Fig3, I am wondering the affect of C105A, C113A mutations on PB1 oligomer formation. To exclude this possibility, the authors should indicate the oligomer formation of p62 mutant by some experiment such as IP, Native-PAGE, or gel filtration chromatography.

We have shown that mutation of C105A,C133A does not affect PB1-dependent aggregate formation. Please see Supplementary Figure S8B,C. We would request not to carry out further biochemical studies since we do not believe they would provide any further novel insight. .

(3) Nrf2 is mainly activated by two distinct pathways, which regulated by p62-dependent and p62-independent system. In normal cells, Nrf2 is usually activated by Keap1-Nrf2 axis respond to oxidative stress. The authors should indicate the regulatory mechanism of Nrf2 activation in the cells producing disulfide-linked p62. It could be easily determined by immunoblotting with anti-phospho p62 (pS349) antibody (Ichimura Y. et al. Mol Cell. 51, 618-631. 2013).

The role of Nrf2 signalling has been raised by the original Reviewers and in our previous rebuttal letter we provided extensive experimental data showing that

disulphide-linked oligomerisation does not affect Nrf2 signalling. Please see below for a thorough discussion of these results.

We investigated the Nrf2 response to H₂O₂ treatment in our cell lines and we do not observe any correlation with increased cell survival of cells expressing wild type vs mutant p62. As can be appreciated in Figure 1 of this document, H₂O₂ treatment leads to a robust upregulation of the Nrf2 pathway indicated by increases in total and nuclear Nrf2 (Fig. 1A, B), as well as induction of Nrf2 target gene *HO-1* (Fig. 1C). We do not observe a significant p62 dependence in response to H₂O₂ in our cells. Most importantly, we do not see significant difference in Nrf2 response between cells expressing wild type and the Cys mutant of p62. Additionally, treatment of cells with retinoic acid, an inhibitor of Nrf2 signalling (Wang *et al.*, 2007), did not negate the rescue of cell death in response to H₂O₂ in cells expressing wild type p62 (Fig. 1D). Collectively, these data do not support that Nrf2 signalling is underlying this mechanism of stress resistance by p62.

Rather, our data demonstrate that upregulation of autophagy acts as the mechanism of resistance to oxidative stress. Inhibition of autophagy using chloroquine or bafilomycin A1 completely cancelled out the difference in the rates of cell survival in response to H₂O₂ treatment between wild type and mutant p62 (Figure 3J). Thus, we believe that our data strongly implicate the upregulation of autophagy by disulphide-linked p62 as a mechanism of increased cell survival due removal of misfolded proteins and reduction of oxidative stress. This conclusion fits well with published studies which demonstrate that turnover of misfolded proteins via autophagy or through the proteasome is sufficient to reduce oxidative stress and promote cell survival (e.g. Cell Rep, 18, 13, p3143–3154, 2017).

Fig. 1

Fig. 1. (A) *p62*^{-/-} MEFs stably expressing FLAG-tagged wild type, C105A,C113A or K7A,D69A PB1 domain mutant *p62* were treated with H₂O₂ (1mM) in serum free media for 5 hours and subjected to a nuclear fractionation followed by immunoblot analysis for Nrf2, Histone 3 and GAPDH and quantified (B). (C) Cells were treated as in (A) and Nrf2 target gene *HO-1* mRNA levels were analysed by qPCR, actin was used as a loading control. (D) *p62*^{-/-} MEFs stably expressing FLAG-tagged wild type or C105A,C113A mutant *p62* were pre-treated with retinoic acid (RA, 30μM) for one hour in serum free media followed by the same treatment as in (A) with or without retinoic acid (30μM) and % cell death was analysed by ReadyProbes fluorescent dyes (Life Technologies).

(4) In this study, autophagy is seemed to be increased depending on the accumulation of disulfide-linked *p62* aggregates as presented in Fig. 3. However, co-localization of *p62* and LC3 signals is not exactly correlated with the activation of autophagy, because *p62* directly binds to LC3 (Pankiv S. et al. J Biol Chem. 282, 24131-24145. 2007). For this, the authors should judge the autophagy activity by the increased number of autophagosomes and autolysosomes with EM analysis.

As requested, we have carried out EM analysis which is now included in Fig. 3G, H. These data support the current data in Fig. 3 showing that autophagosome numbers are increased in cells expressing wild-type *p62* compared to knock-out, oxidation

mutant and PB1 domain mutant. In the same Figure we also show an induction of autophagy (increased LC3-II levels and reduced levels of ubiquitin and p62) by western blot analysis which does not suffer from the limitation highlighted by the reviewer. All together, we believe our data are sufficient to support our original conclusion that disulphide-linked oligomerisation of p62 promotes prosurvival autophagy which correlates well with cell death analysis in oxidative stress conditions (Figure 3I, J).

Reviewer #4 (Remarks to the Author):

I thank the authors for taking care in answering my questions and performing additional experiments to address the criticism from me. By the additional new data, the manuscript has improved somewhat. However, I think that the manuscript is still too preliminary to prove the author's hypothesis.

Regarding the author's comments;

(1) Indeed, we show that PB1 domain-mediated interactions are not required for the formation of disulphide linkages of p62 (Supplementary Figure S5C).

(2) We have shown that mutation of C105A,C133A does not affect PB1-dependent aggregate formation. Please see Supplementary Figure S8B,C.

These results indicate that DLC of C105-C113 of p62 occurs independently of PB1 interaction mediated by K7-D69. Further, PB1 interaction of K7-D69 in p62 can be produced normally even in the DLC-deficient mutant of p62 (C105A, C113A). Nevertheless, the complementary effect between K7A,D69A and C105A,C113A of p62 have not been observed in Figure 3.

In this study, DLC of p62 specifically appears under oxidative stress conditions, it may assist the oligomerization of p62 and autophagy activity, but the physiological significance of p62-oligomer (or p62-aggregate) has been already published. Namely, this manuscript lacks sufficient novelty and impact. Therefore, this reviewer can not support publication of this study on "Nature communications".

Reviewer #5 (Remarks to the Author):

Remarks to the authors

Overall I think the paper opens up a new view point on how p62 could act in response to oxidative stress although mechanistic details remain open. Given the wealth of new cellular data, I think the manuscript warrants publication if the Nrf2 data is included and the detailed interpretation on the side chain level are clearly separated in the discussion and are more clearly treated as speculation.

- Reviewer #4 point 1:

The request of Reviewer #4 to carry out structural studies as done in Ciuffa et al. 2015 appears quite demanding as this may not reflect the expertise of the authors. Although clearly useful suggestions, I agree with the authors to omit such a more comprehensive study.

- Reviewer #4 point 2:

This point is related and I assume Reviewer #4 asks the question from a mechanistic point of view and demands further biophysical or structural characterization of the purified protein to better support the speculative in silico model with experimental data. I believe the proposed analysis would be important to serve as a control whether the C105A or C113A mutants alone already affect oligomer formation. In case the authors are not able to clarify this point, it would be best to remove the details of the in silico model. In particular, the conformation of residues 100 - 120 remains speculative as it was modeled and not validated by the requested in vitro experiments of the Reviewer. The authors can still state this more generally as in their response to Reviewer #4: ... that the two processes of PB1 domain oligomerisation and disulfide linkage can act relatively independently based on the mutant data, however both can contribute to the formation of p62 aggregates. The detailed mechanism and conformation needs to be clarified in future studies as this part of the structure was also not resolved by Ciuffa's cryo-EM study.

- Reviewer #4 point 3:

It is unclear why the authors have not included the Nrf2 data in the manuscript. This is indeed an important point raised and apparently the authors have investigated this point. The authors should add the data to the manuscript provided in the response.

- Reviewer #4 point 4: Done.

Reviewer #5 (Remarks to the Author):

Remarks to the authors

Overall I think the paper opens up a new view point on how p62 could act in response to oxidative stress although mechanistic details remain open. Given the wealth of new cellular data, I think the manuscript warrants publication if the Nrf2 data is included and the detailed interpretation on the side chain level are clearly separated in the discussion and are more clearly treated as speculation.

- Reviewer #4 point 1:

The request of Reviewer #4 to carry out structural studies as done in Ciuffa et al. 2015 appears quite demanding as this may not reflect the expertise of the authors. Although clearly useful suggestions, I agree with the authors to omit such a more comprehensive study.

Response: as suggested by the Editor and the Reviewer we changed the text and figure legend making it clear that the model is a hypothesis that needs to be tested in future structural studies.

- Reviewer #4 point 2:

This point is related and I assume Reviewer #4 asks the question from a mechanistic point of view and demands further biophysical or structural characterization of the purified protein to better support the speculative in silico model with experimental data. I believe the proposed analysis would be important to serve as a control whether the C105A or C113A mutants alone already affect oligomer formation. In case the authors are not able to clarify this point, it would be best to remove the details of the in silico model. In particular, the conformation of residues 100 - 120 remains speculative as it was modeled and not validated by the requested in vitro experiments of the Reviewer. The authors can still state this more generally as in their response to Reviewer #4: ... that the two processes of PB1 domain oligomerisation and disulfide linkage can act relatively independently based on the mutant data, however both can contribute to the formation of p62 aggregates. The detailed mechanism and conformation needs to be clarified in future studies as this part of the structure was also not resolved by Ciuffa's cryo-EM study.

Response: As suggested by the Reviewer we have made the following statement in the Discussion: "Whilst this model requires testing in future structural studies our mutational analyses suggest that the two processes of PB1 domain- and DLC-mediated oligomerisation can act relatively independently, however both can contribute to the formation of p62 aggregates."

- Reviewer #4 point 3:

It is unclear why the authors have not included the Nrf2 data in the manuscript. This is indeed an important point raised and apparently the authors have investigated this point. The authors should add the data to the manuscript provided in the response.

Response: As suggested by the Reviewer the Nrf2 data has been added to the manuscript (new Supplementary Fig. 10).

- Reviewer #4 point 4: Done.